# High resolution cryo-EM and crystallographic snapshots of the actinobacterial two-in-one 2-oxoglutarate dehydrogenase

Lu Yang [1,2], Tristan Wagner[1,4], Ariel Mechaly[3], Alexandra Boyko[1,5], Eduardo M. Bruch[1,6], Daniela Megrian [1], Francesca Gubellini [1], Pedro M. Alzari [1] & Marco Bellinzoni [1] ✉

Actinobacteria possess unique ways to regulate the oxoglutarate metabolic node. Contrary to most organisms in which three enzymes compose the 2-oxoglutarate dehydrogenase complex (ODH), actinobacteria rely on a two-in-one protein (OdhA) in which both the oxidative decarboxylation and succinyl transferase steps are carried out by the same polypeptide. Here we describe high-resolution cryo-EM and crystallographic snapshots of representative enzymes from *Mycobacterium smegmatis* and *Corynebacterium glutamicum*, showing that OdhA is an 800-kDa homohexamer that assembles into a three-blade propeller shape. The obligate trimeric and dimeric states of the acyltransferase and dehydrogenase domains, respectively, are critical for maintaining the overall assembly, where both domains interact via subtle readjustments of their interfaces. Complexes obtained with substrate analogues, reaction products and allosteric regulators illustrate how these domains operate. Furthermore, we provide additional insights into the phosphorylation-dependent regulation of this enzymatic machinery by the signalling protein OdhI.

Acyl-CoA esters are the major metabolic carriers of carbon units in living organisms. The most conserved ways to their synthesis include the oxidative decarboxylation of 2-oxoacids carried out by three main dehydrogenase complexes: the pyruvate dehydrogenase complex (PDHc), that feeds acetyl-CoA units into the TCA cycle, the branched-chain ketoacid dehydrogenase complex (BCKDH) involved in the catabolism of leucine, isoleucine and valine, and the 2-oxoglutarate dehydrogenase complex (ODH; also known as α-ketoacid dehydrogenase, or KDH), which catalyses the synthesis of succinyl-CoA within the TCA cycle[1]. Other substrate specificities have also been reported, one of the most known examples being the 2-oxoadipate dehydrogenase complex which, in human, shares subunits with ODH and is devoted to the degradation of 2-oxoadipate from the L-lysine degradation pathway[2–4]. These enzymatic machineries are made by multiple copies of three enzyme components: the E1 component catalyses the thiamine diphosphate (ThDP)-dependent decarboxylation of the 2-oxoacid and its transfer to a lipoyl-lysine group, E2 transfers this acyl-moiety to the CoASH acceptor to generate acyl-CoA, and E3 is a FAD-dependent dehydrogenase that uses NAD+ to oxidize the dihydrolipoyl moiety, generating NADH[5]. One or more lipoyl domains, which shuttle between the E1, E2 and E3 active sites, are covalently connected to the E2 catalytic domain through flexible linkers acting as swinging arms[1].

[1]Institut Pasteur, Université Paris Cité, CNRS UMR3528, Unité de Microbiologie Structurale, F-75015 Paris, France. [2]Wuhan Institute of Biological Products Co. Ltd., Wuhan 430207, PR China. [3]Institut Pasteur, Université Paris Cité, Plateforme de Cristallographie, F-75015 Paris, France. [4]Present address: Microbial Metabolism Group, Max Planck Institute for Marine Microbiology, Celsiusstraße 1, D-28359 Bremen, Germany. [5]Present address: BostonGene, Yerevan, Armenia. [6]Present address: Sanofi, In vitro Biology, Integrated Drug Discovery, 350 Water St, Cambridge, MA 02141, USA. ✉e-mail: marco.bellinzoni@pasteur.fr

These ubiquitous complexes among aerobic organisms have long been thought to share a universally conserved architecture, characterized by a large hollow central core composed of multiple copies of the E2 catalytic domain, with a symmetry depending on the complex and the species[1,6]. The highly symmetric nature of the E2 core was first described by pioneer electron microscopy and X-ray crystallography studies[7,8], and largely confirmed afterwards[9–12]. The two other complex components, i.e. E1 and E3, are tethered to the core through the peripheral subunit binding domain (PSBD) located on the protruding E2 swinging arms[13,14]. Although such interactions have been characterized and the structure of separate subcomplexes reported[15–20],

## Table 1 | Crystallographic data collection and refinement statistics

| Dataset | *Ms*KGD-GarA | OdhA$_{\Delta97}$ |
|---|---|---|
| Synchrotron beamline | SOLEIL Proxima 1 | ESRF ID30A-3 |
| Wavelength (Å) | 0.9763 | 0.9677 |
| Space group | P 6$_5$ | H 3 2 |
| Unit cell parameters | | |
| *a, b, c* (Å) | 325.75, 325.75, 396.94 | 150.99, 150.99, 314.34 |
| α, β, γ (°) | 90.00, 90.00, 120.00 | 90.00, 90.00, 120.00 |
| Resolution (Å)[a] | 282.11 – 4.56 (4.78 – 4.56) | 100.52 – 2.46 (2.70 – 2.46) |
| $R_{pim}$[b] | 0.076 (0.583) | 0.047 (0.466) |
| *I* /σ(*I*) | 8.1 (1.5) | 15.5 (1.6) |
| Completeness (%) | 94.6 (52.9) | 94.4 (63.8) |
| CC(1/2) | 0.998 (0.701) | 0.998 (0.658) |
| Multiplicity | 11.5 (11.8) | 10.4 (8.5) |
| Refinement | | |
| Resolution (Å) | 4.56 | 2.46 |
| No. reflections | 121081 | 38238 |
| $R_{work}$/ $R_{free}$ (%)[c] | 19.8 / 22.9 | 20.4 / 25.1 |
| No. atoms | | |
| Protein | 56835 | 8366 |
| Ligands/ions | 170 | 109 |
| Solvent | – | 397 |
| Average B-factors | | |
| Protein | 236.15 | 59.22 |
| Ligand/ions | 222.98 | 76.32 |
| Solvent | – | 49.85 |
| R.m.s deviations[d] | | |
| Bond lengths (Å) | 0.009 | 0.011 |
| Bond angles (°) | 1.278 | 1.458 |
| Validation[d] | | |
| MolProbity score | 2.40 | 1.20 |
| Clashscore | 8.15 | 1.56 |
| Poor rotamers (%) | 4.56 | 1.71 |
| Ramachandran plot[d] | | |
| Favored (%) | 92.72 | 97.41 |
| Allowed (%) | 6.94 | 2.59 |
| Outliers (%) | 0.34 | 0.00 |
| PDB accession code | 8P5R | 8P5S |

[a]Resolution limits were determined by applying an anisotropic high-resolution cut-off via STARANISO, part of the autoPROC data processing software[54]; values in parentheses refer to the highest resolution shell.
[b]$R_{pim} = \Sigma_{hkl}[1/(N-1)]^{1/2}\Sigma_i|I_i(hkl) - \langle I \rangle(hkl)|/\Sigma_{hkl} \Sigma_i I_i(hkl)$, where N is the multiplicity, $I_i$ is the intensity of reflection i and $\langle I \rangle(hkl)$ is the mean intensity of all symmetry-related reflections.
[c]$R_{work} = \Sigma||F_o| - |F_c||/\Sigma|F_o|$, where $F_o$ and $F_c$ are the observed and calculated structure factor amplitudes. Five percent of the reflections were reserved for the calculation of $R_{free}$.
[d]Values from MOLPROBITY[58].

the intrinsic flexible nature of the E2 swinging arms has long hampered attempts to perform high-resolution structural studies of these complexes in their entirety. Progress has been made recently, with single particle cryo-EM studies that have shed light on dihydrolipoyl-lysine channeling within the E2 core of *E. coli* PDHc[21], among the most studied models for such complexes, as well as on the role of the E3 binding protein (E3BP) that is also part of the inner core in eukaryotic PDH complexes[22–24].

These highly conserved principles of 2-oxoacid dehydrogenase assembly, however, are not followed in Actinobacteria, one of the largest bacteria phyla. Earlier work had shown that *Corynebacterium glutamicum*, a well-known actinobacterial model largely used for biotechnological applications, possesses an enzyme, called OdhA, which bears succinyltransferase (E2o) and 2-oxoglutarate decarboxylase (E1o) domains on the same polypeptide[25–27], a feature shared by the mycobacterial homolog KGD[28]. Considering the presence of such a 'two-in-one' fusion enzyme, which depends on lipoyl-lysine provided in trans as an acyl group carrier, and that E2p was reported to be the only lipoylated protein in *M. tuberculosis*[29] and in *C. glutamicum*[26], it was proposed that PDH and ODH may form a mixed supercomplex in those species, a hypothesis corroborated by the copurification of OdhA with components of the PDH complex in *C. glutamicum*[27]. The three-dimensional architecture of OdhA has however remained unknown so far, raising questions not only on how domains characterized by different oligomeric states may be arranged in the same polypeptide, but also about the composition, the size and the assembly of such a mixed PDH/ODH supercomplex. In addition, challenging even further the current 'dogma' of the universal conservation of PDH and ODH complexes, we recently showed that AceF (E2p) in actinobacterial PDH is reduced to its minimally active trimeric unit, due to a three-residue insertion at its C-terminal end that hinders any trimer-trimer interaction[30]. We also proposed that the presence of the C-terminal insertion and that of an *odhA*-like gene are related and constitute a signature of the Actinobacteria class[30].

Here, we show by X-ray crystallography and high-resolution cryo-EM, that corynebacterial OdhA and its mycobacterial orthologue KGD are large (~0.8 MDa) homohexameric enzymes with an unprecedented molecular architecture, and discuss how intra and interdomain interactions may account for their unusual regulatory properties.

## Results

### Actinobacterial OdhA/KGD is an 800 kDa homohexamer with two distinct catalytic centers

Our previous work with an N-terminal truncated form of *Mycobacterium smegmatis* KGD (*Ms*KGD$_{\Delta115}$ construct designed based on limited trypsin proteolysis of the full-length protein) revealed the structure of the active E1o homodimer, in which each protomer was tightly associated to a monomeric E2o-like domain[28]. Since the acyltransferase catalytic activity is located at the junction of two E2 protomers[31], the obtained structure cannot reflect an active E2o state, and we hypothesized that the N-terminal truncation may have interfered with the assembly of the obligate homotrimeric state of this domain. We therefore decided to produce in *E. coli* the full-length proteins KGD from *M. smegmatis* and OdhA from *C. glutamicum* for further studies. Crystals were obtained for both proteins, but their X-ray diffraction was too limited for structural characterization. Therefore, we first proceeded by co-crystallizing *Ms*KGD with the inhibitor GarA for further stabilization, and we solved the structure of the *Ms*KGD-GarA complex at 4.6 Å resolution using the previously published *Ms*KGD$_{\Delta115}$ structure[28] as the search model (Table 1). *Ms*KGD presents an homohexameric assembly (Fig. 1), which can be described as a three-blade, triangular propeller shape, approximately 20 nm wide and 15 nm thick. The E1o dimers containing the Mg-ThDP compose the blades, with two central E2o trimers sitting respectively on either side of the blades plane. This oligomeric arrangement as a trimer of dimers allows the

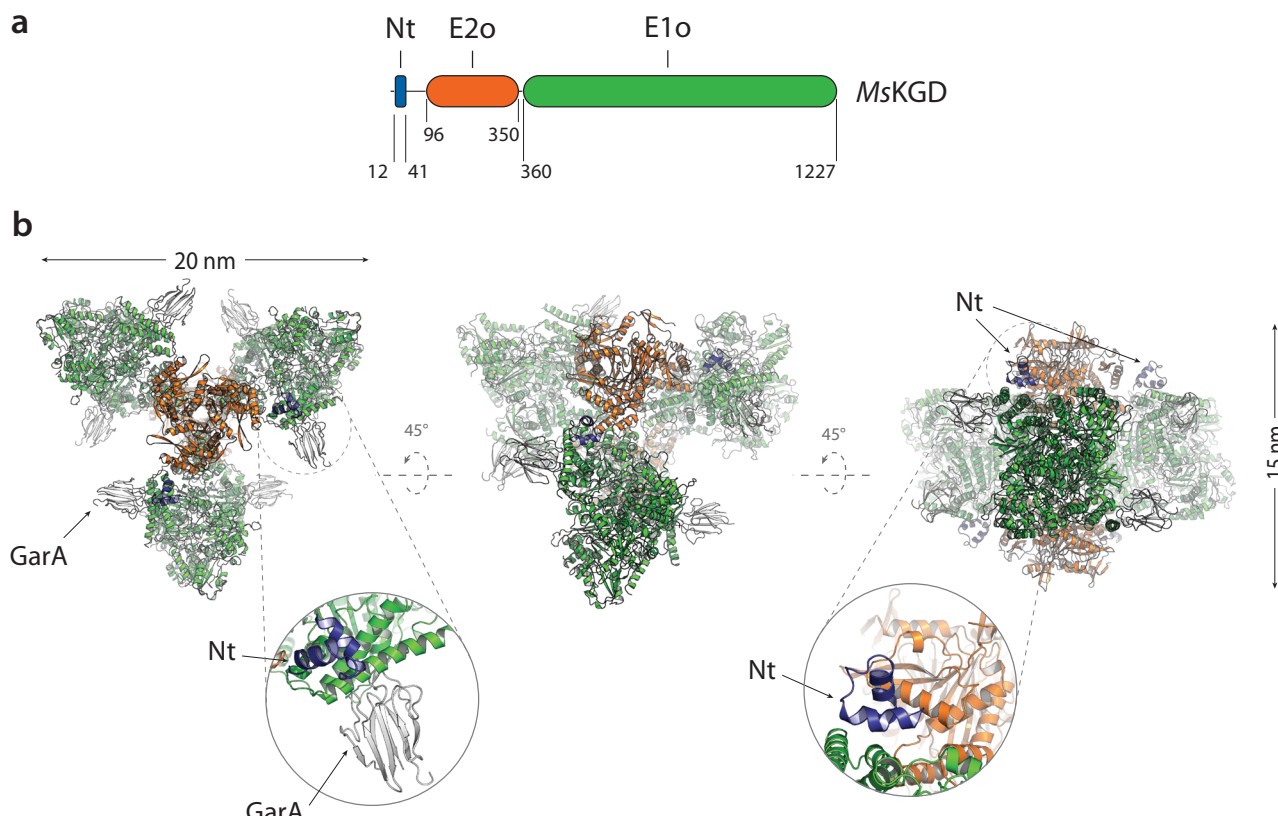

**Fig. 1 | *Ms*KGD domain boundaries and crystal structure of the *Ms*KGD-GarA complex. a** Domain boundaries in *Ms*KGD (Nt: N-terminal helical domain) and (**b**) cartoon overview of the *Ms*KGD hexamer in complex with GarA (gray). Zoomed views highlighting the position of the N-terminal helical hairpin (blue) are shown below. Crystal structure determined at 4.6 Å resolution.

separate E1o and E2o domains to maintain their canonical oligomeric arrangements (i.e. E1o dimers and E2o trimers) as seen in other oxoacid dehydrogenases, with their functional catalytic centers at the respective oligomeric interfaces. The inhibitor GarA binds full-length *Ms*KGD in the same way as previously described for the high-resolution structure of the GarA-*Ms*KGD$_{\Delta360}$ complex[32] (Supplementary Fig. 1), and the *Ms*KGD E1o domain is indeed held in the resting conformation, as a result of GarA binding[32]. Interestingly, the low-resolution crystal structure also shows clear electron density, in four out of the six protomers, for a short helical hairpin engaged in intermolecular interactions (Fig. 1b), which could be attributed to the N-terminal helical segment[26,28]. These observations suggest that the N-terminal domain of *Ms*KGD and OdhA could be involved in protein-protein interactions with other components of the complex, as it was recently reported for human[33] and bovine ODH[34].

In parallel, the characterization of recombinant C. *glutamicum* OdhA showed a specific 2-oxoglutarate decarboxylase activity of 110.3 ± 1.0 nmol/min/mg, consistent with previous reports (Supplementary Table 1)[26], and an ODH activity of 68.6 ± 0.6 nmol/min/mg when integrated in a reconstituted PDH/ODH supercomplex with recombinant AceE (E1p), AceF (E2p) and Lpd (E3) from the same species in equimolar ratios, a value higher than reported elsewhere under similar conditions[35] but compatible with measurements made on C. *glutamicum* cell extracts (Supplementary Table 1). The same reconstituted supercomplex showed a lower PDH specific activity (3.3 ± 0.3 nmol/min/mg), which however raised to 14.5 ± 0.4 nmol/min/mg when OdhA was omitted (Supplementary Table 1), suggesting that OdhA and AceE may compete for the available lipoyl groups, consistently with AceF being required for both the PDH and ODH reactions[26]. On the other hand, purified OdhA showed a sedimentation coefficient of 16.8 S (Supplementary Fig. 2) suggesting a predominant homohexameric state in solution, at all the tested concentrations. Therefore, based on the *Ms*KGD structure and on secondary structure predictions, we produced a truncated version of OdhA deprived of the flexible N-terminal segment (OdhA$_{\Delta97}$), which produced crystals diffracting up to 2.5 Å resolution (Table 1). As OdhA indeed presents the same homohexameric assembly as *Ms*KGD (Fig. 2), all further structural analyses in this work will be focused on the higher resolution OdhA model. The single protomer shows the presence of an N-terminal acyltransferase (E2o) domain, spanning from the construct N-terminus to residue Asn349, connected by a 17-residue linker to the C-terminal, ThDP-dependent oxoglutarate dehydrogenase domain (E1o) (Supplementary Fig. 3). In turn, the latter is made by a small helical domain (residues Asp367-Thr448) followed by three consecutive α/β subdomains, characteristic of homodimeric transketolases (Supplementary Fig. 3). In addition, a ThDP-Mg$^{2+}$ cofactor is bound at the E1o domain dimeric interface (Fig. 2a), in an equivalent pose and active site environment as in the *M. smegmatis* E1o (*Ms*KGD$_{\Delta360}$) high-resolution crystal structures[28,36]. On the other hand, the E2o N-terminal domain shows the known, compact triangular trimeric conformation of the chloramphenicol acetyltransferase (CAT) family, characterized by an N-terminal β-strand that protrudes to make a strand exchange (mixed β-sheet) with the neighboring monomer (Supplementary Fig. 4). The absence of such β-strand and the following α-helix in the *Ms*KGD$_{\Delta115}$ construct may therefore explain initial failures in observing a functional E2o assembly[28]. Furthermore, despite the overall conservation of the CAT fold, a notable difference between the OdhA succinyltransferase domain and other E2o domains resides in the β-harpin that normally lies close to the three-fold axis and is located on the inside surface of the cubic assembly, here replaced by a short α-helix

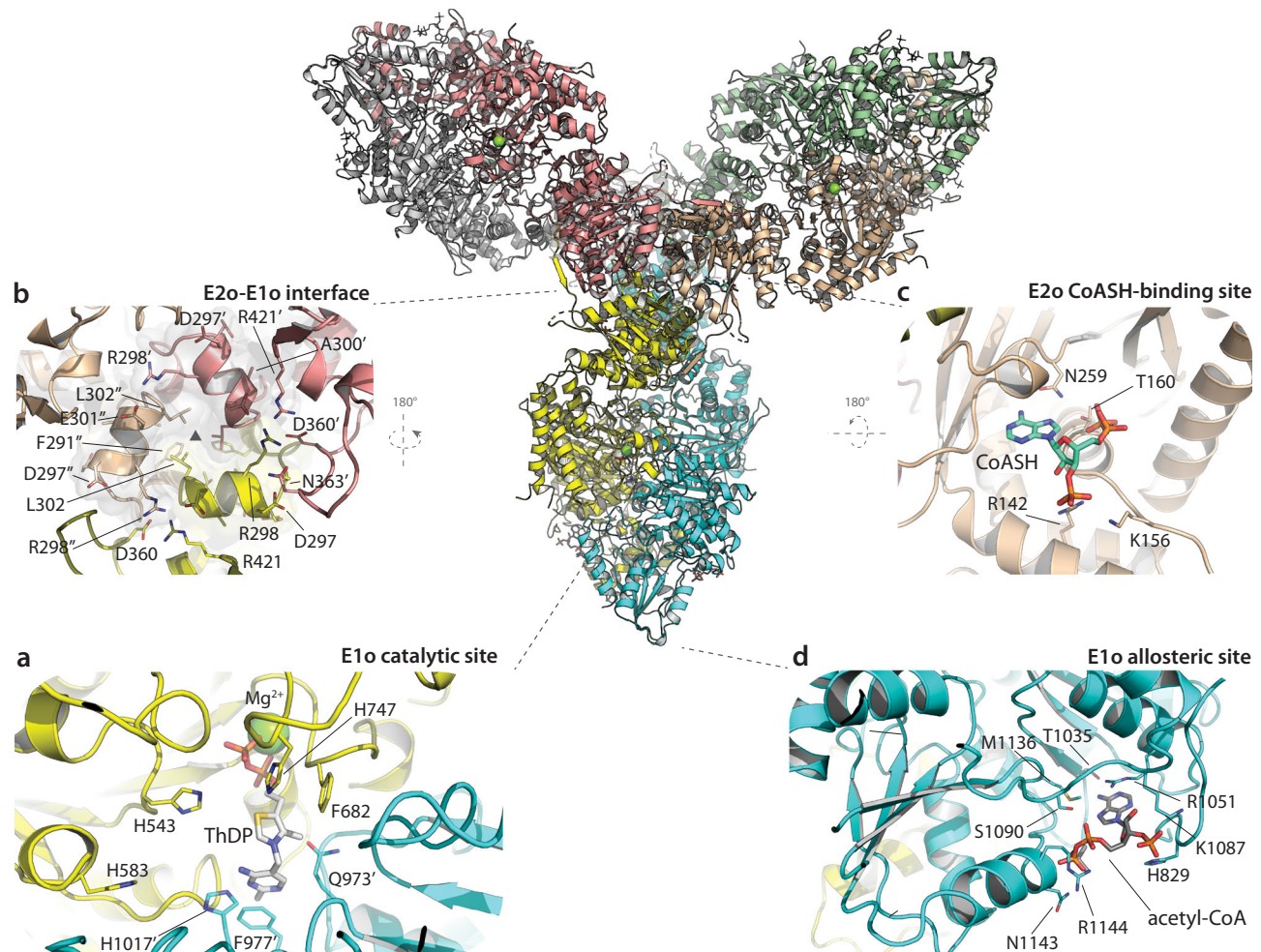

**Fig. 2 | Crystal structure of OdhA$_{\Delta 97}$ with focus on the E2o/E1o interface and on catalytic and allosteric sites.** Center: cartoon overview of the OdhA$_{\Delta 97}$ homo-hexamer (one color per chain), with the single protomers related by crystal-lographic symmetry. Laterally, clockwise: **a** E1o catalytic site at the protomer interface, the ThDP-Mg$^{2+}$ cofactor at the center; **b** view, from the inside face, of the E2o-E1o interface along the E2o domain 3-fold axis (triangle), highlighting the symmetric intra- and intersubunit interactions of the short α-helix Glu296-Leu302 (see also Supplementary Fig. 4); **c** E2o CoASH binding site (CoASH pantothenate chain not traced due to lack of supporting electron density); **d** E1o allosteric acetyl-CoA binding site (pantothenate chain also not traceable). Indicated in the figure and depicted as sticks are residue side chains interacting with cofactor or ligands (**a, c, d**), involved in contacts at the domain interface (**b**) or with a predicted role in catalysis (namely His543, His583, His747 and His1017, as reported for *Ms*KGD[28]).

(connected by flexible linkers) spanning residues 296-302 (Fig. 2b; Supplementary Fig. 4). This helix interacts both with its symmetric counterparts from the neighboring domains around the E2o three-fold axis, notably through a strong salt bridge involving Glu301 and Arg298, but also makes contacts with the E1o domains (Fig. 2b), both intrasu-bunit (the Ala300 carbonyl oxygen is well positioned to hydrogen bond to Arg421) and intersubunit, another H-bond involving the car-boxyl group of Asp297 and the main chain amide of Asn363 from the nearby chain (Fig. 2b). The three-fold symmetric packing of the 296-302 α-helix is further stabilized by hydrophobic interactions between the side chains of Phe291 and Leu302, positioned internally (Fig. 2b). The structural alignment of the OdhA and *Ms*KGD E2o domains with characterized acyltransferase domains from other E2 enzymes con-firms that this α-helix arises from a sequence insertion (Supplementary Fig. 5), and suggest it to be a structural feature of OdhA-like enzymes, likely as an adaptation to E2o-E1o fusions. As a result of this hexameric arrangement, the E1o and E2o active sites are poised at approximately 60 Å one to the other, with the oxoacid substrate and the acceptor CoASH getting access to them from different sides of the propeller 'blade' (Supplementary Fig. 6). Consistently, we could model CoASH, which was added to the cocrystallization mixture, as bound to the E2o

acceptor site (Fig. 2c), with the adenosine moiety adopting an equivalent pose to previously reported complexes with E2 enzymes[30,31].

Further inspection of the electron density maps revealed the presence of another 3'-phosphonucleotide, bound to the E1o domain in a pocket that was previously identified as the allosteric acetyl-CoA site in mycobacterial KGD[28] (Fig. 2d). We therefore modeled this ligand as acetyl-CoA, noting that supporting electron density for the pan-tothenate chain was also absent in *Ms*KGD$_{\Delta 360}$ when crystals were soaked with millimolar concentrations of acetyl-CoA, in the absence of the 2-oxoglutarate substrate (PDB 2XTA; Supplementary Fig. 7)[28], suggesting a shared regulation mechanism between the two enzymes.

## High-resolution cryo-EM studies of OdhA

To study the conformational changes triggered by substrates or allosteric regulators that might affect the domain reorganization, high-resolution single particle cryo-EM was employed for further structural characterization. After assessing the suitability of full-length OdhA samples for single particle analysis using negative staining EM (Sup-plementary Fig. 8), plunge-frozen samples were prepared at different protein concentrations in the presence of the oxoglutarate analog

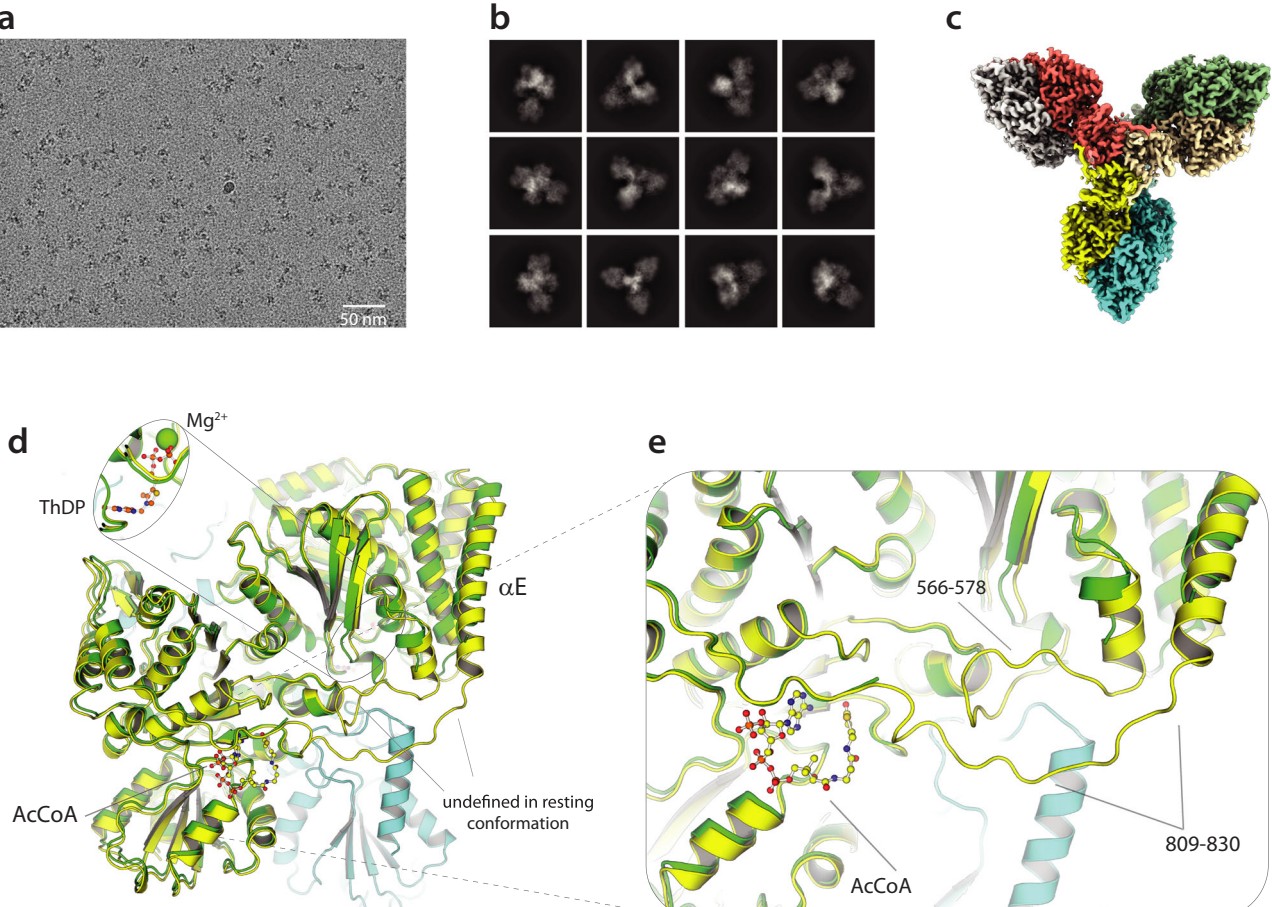

**Fig. 3 | High-resolution single particle cryo-EM structure determination of OdhA. a** Representative micrograph of an OdhA sample vitrified on an UltraAuFoil grid (Quantifoil), following motion correction and CTF estimation.
**b** Representative 2D classes from the same dataset as in **a**. **c** Overall representation of the OdhA EM map at 2.2 Å resolution, showing the OdhA homohexamer (one color per chain). **d** Superimposition, focused on the E1o domain, of the OdhA$_{\Delta 97}$ model (green) to the OdhA model determined by single particle cryo-EM in the absence of added ligands (yellow/light blue). Loops that could not be traced in OdhA$_{\Delta 97}$ are indicated. Also, to be noted the different position of the αE external helix (see also Supplementary Fig. 12). **e** Zoomed view on the region most

concerned by the conformational change, from the resting conformation (observed for the OdhA$_{\Delta 97}$ crystal structure) to the active conformation observed by cryo-EM. Loops that become well defined in the active conformation are indicated, as well as the acetyl-CoA molecule observed in the allosteric E1o pocket in the OdhA model obtained by cryo-EM. An acetyl-CoA molecule bound in the same pocket was observed in all the OdhA complexes solved by single particle cryo-EM, except for the OdhA-OdhI complex. The OdhA$_{\Delta 97}$ model also presents a CoA ester (presumably acetyl-CoA) bound in the same pocket, although the lack of supporting electron density hindered tracing of the whole pantothenate chain (Fig. 2).

succinyl phosphonate (SP), previously shown to stabilize KGD[37]. Single particle cryo-EM allowed us to get a first OdhA map at 3.4 Å resolution, following ab initio reconstruction and 3D refinement applying dihedral D3 symmetry. However, the narrow particle distribution precluded us to improve the map resolution. Raising OdhA concentration up to about 8 mg/ml and including 8 mM CHAPSO in the sample before plunge-freezing allowed a significant improvement of both the number of particles per micrograph and their orientation distribution (Fig. 3a/b), as reported in other cases[38], increasing the resolution of the reconstructed map up to 2.3 Å (Supplementary Fig. 9). A further improvement of the map up to 2.2 Å (Fig. 3c) was obtained by combining maps generated by local refinement of the two separate domains (Supplementary Fig. 9; Supplementary Movie 1). The same grid preparation strategy was then applied to other full-length OdhA samples, i.e. enzyme without added ligands, or preincubated with either CoASH or succinyl-CoA, leading to maps at comparable resolutions of 2.1-2.2 Å (Supplementary Fig. 10; Table 2). In all cases, the maps, which showed well-defined density for most side chains, allowed to trace the OdhA polypeptide chain unambiguously starting from residue Pro102, corresponding to the N-terminal boundary of the E2o domain, with excellent stereochemical parameters (Table 2). In

contrast, the full N-terminal OdhA segment, corresponding to the first hundred residues that include a predicted helical hairpin analogous to the one observed in the crystal structure of MsKGD (Fig. 1b), could not be traced due to the lack of supporting density, confirming its high mobility in solution. The E1o active site at the dimer interface showed a clear density for ThDP-Mg²⁺ in all cases (Supplementary Fig. 11). In the case of the OdhA-SP complex, the phosphonate molecule, determined by surface plamon resonance to bind OdhA with a $K_D$ of $119 \pm 17\ \mu M$, could be modeled as covalently linked to the reactive C2 carbon from the ThDP thiazolium ring (Supplementary Fig. 12). Such adduct, equivalent to the one generated upon cocrystallization of MsKGD$_{\Delta 360}$ (PDB 6R29[37]), provides an excellent mimic of the pre-decarboxylation complex and, in turn, of the incoming 2-oxoglutarate substrate (Supplementary Fig. 12).

Previous work on MsKGD has shown the existence of two different conformational states of its E1o domain, i.e. a resting (or *early*) state *vs.* an activated (or *late*) state[28,36]. The activated state was trapped following the addition of substrates and was associated to post-decarboxylation ThDP-bound intermediates deriving from either 2-oxoglutarate or 2-oxoadipate[36], or phosphonate analogs[37]. In contrast to the crystallographic structure of OdhA$_{\Delta 97}$ which fits the resting

**Table 2 | Cryo-EM data collection, refinement and validation statistics**

| Dataset | OdhA | OdhA-CoASH | OdhA-succinyl-CoA | OdhA-SP | OdhA-OdhI |
|---|---|---|---|---|---|
| Data collection and processing | | | | | |
| Grid type | UltrAuFoil300 mesh R1.2/1.3 | UltrAuFoil300 mesh R1.2/1.3 | UltrAuFoil300 mesh R1.2/1.3 | Lacey 200mesh | Lacey 200mesh |
| Plunge freezer | Vitrobot | Vitrobot | Vitrobot | Vitrobot | Vitrobot |
| Microscope | Krios | Krios | Krios | Krios | Krios |
| Magnification | 105000 | 105000 | 105000 | 105000 | 105000 |
| Voltage (kV) | 300 | 300 | 300 | 300 | 300 |
| Energy filter(eV) | 20 | 20 | 20 | 20 | 20 |
| Camera | K3 | K3 | K3 | K3 | K3 |
| Detector mode | Counted | Counted | Super-resolution | Counted | Counted |
| Electron exposure (e–/Å$^2$) | 40 | 40 | 48 | 40 | 40 |
| Defocus range (μm) | −0.8 to −2.0 | −0.8 to −2.0 | −0.8 to −2.0 | −0.8 to −2.2 | −0.8 to −2.2 |
| Pixel size (Å) | 0.86 | 0.86 | 0.84 | 0.86 | 0.86 |
| Micrographs | 13348 | 12202 | 11827 | 16647 | 19443 |
| No. of fractions | 40 | 40 | 50 | 60 | 40 |
| Symmetry imposed | D3 | D3 | D3 | D3 | D3 |
| Initial particle images (no.) | 6571029 | 3392043 | 3827956 | 2599031 | 3177955 |
| Final particle images (no.) | 1474608 | 1849406 | 1074837 | 646352 | 958690 |
| Map resolution (Å)[a] | 2.17 | 2.17 | 2.07 | 2.26 | 2.29 |
| FSC threshold | 0.143 | 0.143 | 0.143 | 0.143 | 0.143 |
| Map resolution range (Å) | 1.9–4.4 | 1.9–4.5 | 1.9–4.7 | 2.0–4.6 | 2.0–6.0 |
| Refinement and validation | | | | | |
| Model-map resolution (Å)[b] | 2.3 | 2.2 | 2.2 | 2.3 | 2.3 |
| FSC threshold | 0.5 | 0.5 | 0.5 | 0.5 | 0.5 |
| Map sharpening B factor (Å$^2$) | −84.4 | −88.5 | −74.2 | −87.3 | −93.3 |
| Model composition | | | | | |
| No. Atoms (non-H) | 52590 | 52878 | 54248 | 52656 | 55346 |
| Protein residues | 6714 | 6714 | 6726 | 6714 | 7110 |
| Ligands | 18 | 24 | 24 | 18 | 12 |
| Water molecules | – | – | 1280 | – | – |
| Average B factors (Å$^2$) | | | | | |
| Protein | 45.28 | 43.41 | 67.11 | 45.07 | 47.22 |
| Ligands | 56.18 | 57.52 | 83.72 | 56.28 | 123.89 |
| R.m.s. deviations[b] | | | | | |
| Bond lengths (Å) | 0.010 | 0.010 | 0.006 | 0.008 | 0.013 |
| Bond angles (°) | 1.247 | 1.229 | 1.032 | 1.121 | 1.810 |
| Validation[b] | | | | | |
| MolProbity score | 1.00 | 1.02 | 1.08 | 0.88 | 0.90 |
| Clashscore | 1.05 | 1.25 | 1.82 | 0.77 | 0.66 |
| Poor rotamers (%) | 0.42 | 0.31 | 0.53 | 0.11 | 0.21 |
| Ramachandran plot[b] | | | | | |
| Favored (%) | 96.82 | 97.02 | 97.26 | 97.24 | 96.87 |
| Allowed (%) | 3.18 | 2.97 | 2.74 | 2.67 | 3.13 |
| Outliers (%) | 0.00 | 0.01 | 0.00 | 0.09 | 0.00 |
| PDB accession code | 8P5T | 8P5U | 8P5V | 8P5W | 8P5X |
| EMDB accession code | EMD-17452 | EMD-17453 | EMD-17454 | EMD-17455 | EMD-17456 |

[a]Resolution estimates from cryoSPARC (version v3.2.0)[61].
[b]Values from MOLPROBITY[58] and the PHENIX[59] EM validation tools.

state, all our models refined on single particle EM maps adopt an activated conformational state of the E1o domain, even in the absence of added ligands. The state is indeed revealed by the shifts in the loops 566-579 as well as 809-836, that could be traced in the EM structures but were mostly unstructured in the OdhA$_{Δ97}$ crystal structure (Fig. 3d/e; Supplementary Fig. 13)[28,36]. However, all the OdhA cryo-EM datasets, including those corresponding to complexes with CoASH, succinyl-CoA and SP, show an acetyl-CoA molecule bound to E1o allosteric site which could be positioned unambiguously (Fig. 3d/e; Supplementary Fig. 14). The presence of a bound acetyl-CoA activator, most likely acquired following heterologous overexpression of the enzyme in *E. coli*, may therefore explain the observed OdhA activated conformation through a mechanism that involves the stabilization of the loop 809-836 in its extended form (Supplementary Fig. 14), promoting, in turn,

**a**

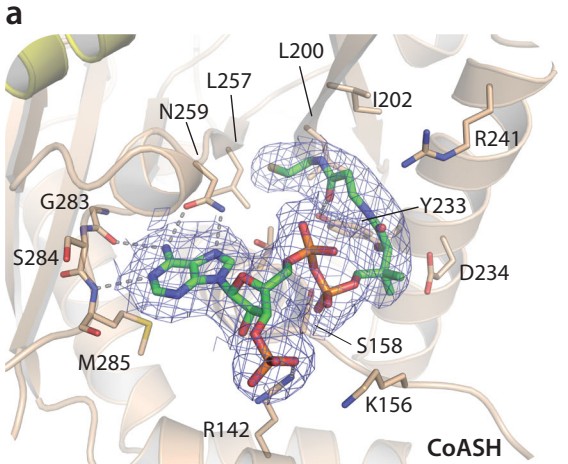
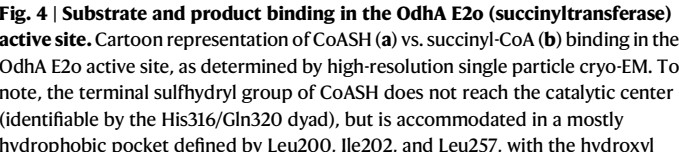

**b**

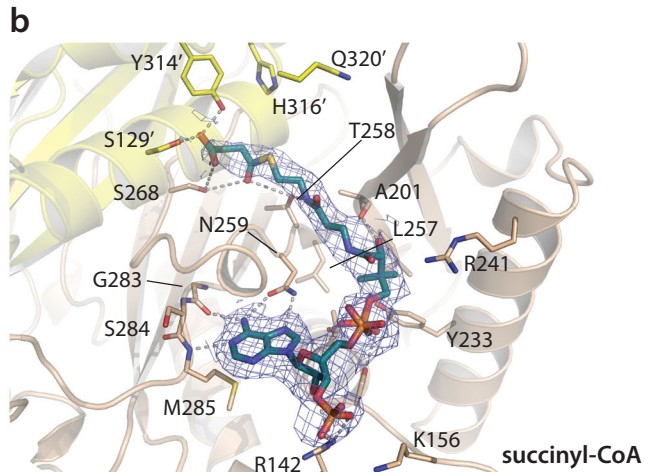

**Fig. 4 | Substrate and product binding in the OdhA E2o (succinyltransferase) active site.** Cartoon representation of CoASH (**a**) vs. succinyl-CoA (**b**) binding in the OdhA E2o active site, as determined by high-resolution single particle cryo-EM. To note, the terminal sulfhydryl group of CoASH does not reach the catalytic center (identifiable by the His316/Gln320 dyad), but is accommodated in a mostly hydrophobic pocket defined by Leu200, Ile202, and Leu257, with the hydroxyl group of Tyr233 acting as a hydrogen bond donor to the terminal carbonyl oxygen of CoASH. This binding orientation, although not identical, corresponds to the 'OUT' conformation originally observed in the ternary complex of *A. vinelandii* E2p with CoASH and free lipoamide (PDB 1EAB; Supplementary Fig. 15). Blue meshes corresponds to the EM map for the ligands, contoured at the 3.5σ level.

the activated conformation[28]. This hypothesis is supported by previous observations showing that acetyl-CoA binding to *Ms*KGD contributed to stabilize the activated conformation[28], as well as by steady-state kinetic and spectroscopy studies that concluded that acetyl-CoA acts as a mixed V and K type allosteric activator on mycobacterial KGD[39].

The ensemble of high-resolution OdhA single-particle cryo-EM complexes provides insights into the functionality of the E2o succinyltransferase domain. First, the OdhA-CoASH complex shows the bound CoASH with the pantothenate chain not entering the active site, but with the terminal, reactive sulphydryl group accommodated in the mostly hydrophobic pocket defined by Leu200, Ala201, Thr258 and the side chains of Ile202 and Leu257 (Fig. 4a). Such CoASH binding mode is close to the previously reported 'out' conformation of CoASH in the non-proficient, ternary complex of *Azotobacter vinelandii* E2p (PDB 1EAB) (Supplementary Fig. 15), where it was proposed as a mechanism to protect the reactive sulfhydryl group from oxidation[31]. In contrast, succinyl-CoA binds to the same domain with its 2-phosphoadenosine moiety superimposable to the one observed for CoASH, but the pantetheine arm directed towards the E2o active site. Noteworthy, its pose is overall very close to the one shown by CoASH in its ternary complex with lipoamide in AceF (E2p) from *C. glutamicum*[30] (Supplementary Fig. 16). Specifically, the sulfur atom is positioned at 5 Å from the NE2 nitrogen of the catalytic His316 belonging to the neighboring subunit (Fig. 4b), a distance compatible with the proposed catalytic mechanism[31], while the terminal carboxyl group from the succinyl moiety is stabilized by hydrogen bonds with Ser129 and Tyr314, also provided by the adjacent subunit. At the same time, the ketone oxygen acts as H-bond acceptor to Thr258 and Ser268 (Fig. 4b; Supplementary Fig. 15/16). The observed succinyl-CoA binding mode agrees with mutagenesis data pointing to a catalytic role for His316 and Gln320, and suggesting Thr258 as involved in CoA binding (Supplementary Fig. 16)[26]. It is worth noting that both Ser129 and Tyr314 are conserved among OdhA orthologues (Supplementary Fig. 17) as well as in structurally characterized E2o enzymes, but not in E2s with different substrate specificity (Supplementary Fig. 5), consistently with the observed role of these residues in stabilizing the terminal carboxyl group from the succinyl moiety. Our snapshots therefore suggest them as one of

the structural features that may contribute to provide substrate selectivity to E2 enzymes.

## FHA regulation: specific interactions for a conserved inhibition mechanism

By preincubating purified OdhA with an excess OdhI and passing the sample through size exclusion chromatography prior to grid preparation, we were also able to obtain a single particle reconstruction of a full OdhA-OdhI complex at 2.3 Å resolution (Supplementary Movie 2). The overall structure is very similar to that of the homologous *Ms*KGD-GarA complex (Figs. 1 and 5a), with OdhI molecules binding, through their FHA domains (traceable for residues Glu40-Ala142), to the OdhA E1o domain with a 1:1 stoichiometry. In contrast to the other single particle EM structures, OdhA adopts here the resting conformation, equivalent to the one observed in the crystal structure of OdhA$_{Δ97}$ and consistently with structural and kinetic observations on mycobacterial KGD, both indicating that GarA binding stabilized this enzyme conformation[32,39]. Moreover, no ligand bound to the OdhA acetyl-CoA allosteric site could be detected in this complex, suggesting that bound acetyl-CoA may have been lost because of OdhI binding, further confirming the link between the presence of the activator and the conformational state of the enzyme.

Interactions with OdhA involve the tips of both OdhI FHA antiparallel β-sheets: one anchors firmly to the OdhA loop Leu591-Glu598, which connects two antiparallel β-strands, while the tip of the other OdhI β-sheet binds the OdhA α-helices Gln480-Lys503 and Asn786-Asn805 (αE) (Fig. 5b). The interactions of the former involve the OdhI positively charged Arg53, Lys132 and Arg134 side chains which bind, through a network of hydrogen bonds, to main chain carbonyl oxygens of the OdhA 591-598 loop (Fig. 5b), in a similar way as in the *Ms*KGD-GarA complex[32]. However, the interactions between OdhI and the OdhA helix αE, which is a landmark of the enzyme activation state[28,36], show a few significant differences when compared to the mycobacterial complex. The αE helix could only be traced till residue 805 in OdhA, and it shows a 30° kink towards OdhI at its N-terminal tip (Fig. 5c). Most notably, no hydrogen bond was observed between the phosphomimetic residue Asp795 in OdhA and OdhI Ser86, in contrast to structural observations on the GarA-*Ms*KGD complex[32,40] but consistently with site-directed mutagenesis on OdhI, which pointed to

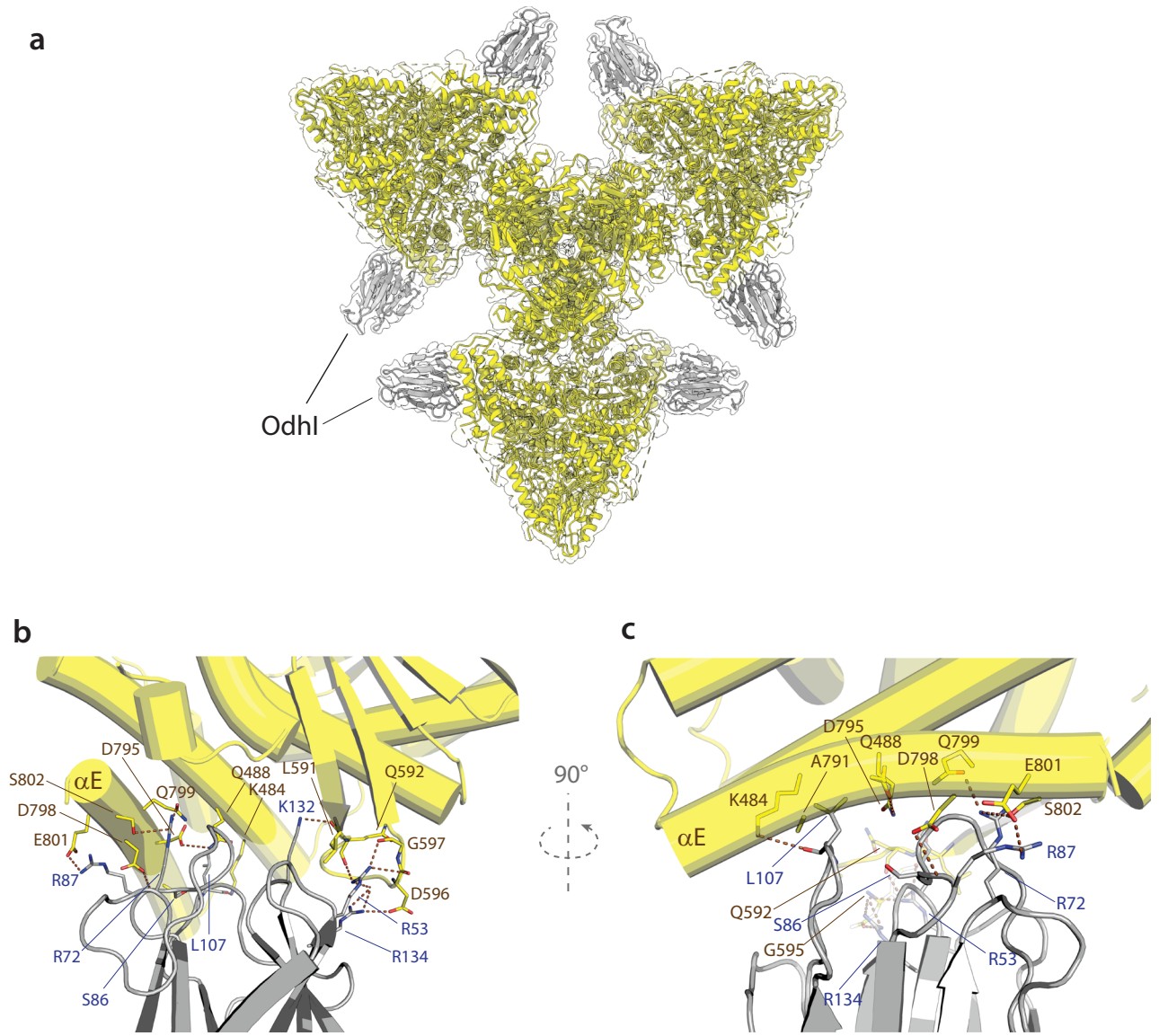

**Fig. 5 | Single particle cryo-EM structure of the OdhA-OdhI complex.**
**a** Visualization of the OdhA-OdhI complex fit in the corresponding single particle EM map at 2.3 Å resolution. **b** Detailed view of the interactions between OdhA (yellow) and OdhI FHA domain (gray), with involved residues depicted as sticks.

Dashed lines indicate hydrogen bonds and salt bridges. OdhA helices are depicted as cylinders. **c** Rotated view of the OdhA-OdhI interactions. αE refers to the OdhA α-helix Ser785-Asn805, following the original *Ms*KGD nomenclature.

Ser86 as dispensable for binding[41]. Accordingly, the binding affinity of the GarA S95A variant for *Ms*KGD was comparable to the wild-type[32].

A negatively charged side chain from Asp798, adjacent to Asp795 on the αE outside surface (instead of a glycine in *Ms*KGD and other orthologues; Supplementary Fig. 17), exists additionally in OdhA within hydrogen-bonding distance to the main chain amino group of OdhI Arg87. The side chain of the same Arg87, in turn, is involved in a salt bridge with OdhA Glu801 (Fig. 5b/c), pointing to a key role of this residue in the OdhI-OdhA interaction. In agreement with these observations, mutations leading to the substitution of OdhI Arg87 to either proline or alanine have been isolated in suppressor mutants of a *glnX* gene deletion in *C. glutamicum*, where the impaired OdhA inhibition overcomes the accumulation of unphosphorylated OdhI[42]. In the same work, a missense mutation involving OdhI Leu107 was also isolated in a mutant strain bearing the same suppressor phenotype[42], consistently with this residue being located at the OdhA-OdhI interface. Leu107 is indeed involved in van der Waals interactions with OdhA Ala791 and the side chains of both Lys484 and Gln488 (Fig. 5c; Supplementary Fig. 18), the substitution of which was shown to impair

the *Ms*KGD-GarA interaction[32]. Overall, the OdhI relative position is shifted approximately 2 Å aside from OdhA when compared to GarA in the corresponding mycobacterial *Ms*KGD-GarA complex (distance calculated as the RMSD over the ensemble of Cα; Supplementary Fig. 18), resulting in a ~5 Å distance between the OdhI Ser86 OG oxygen and the carboxyl group of OdhA Asp795. An additional, distinct intermolecular interaction in OdhA-OdhI is due to the presence of a serine residue at OdhA position 802, still positioned on the αE helix and making a hydrogen bond to OdhI Arg72 (Fig. 5b), while an arginine is found at the corresponding position in *Ms*KGD, as well as in other OdhA-like enzymes (Supplementary Fig. 17). Replacing this residue by an alanine was indeed found to decrease 6.25-fold the $K_i$ of GarA for KGD[32]. Overall, despite a similar molecular surface occluded on the FHA domain upon the interaction with either *Ms*KGD or OdhA (around 950 Å²), and a conserved inhibition mechanism, our high resolution cryo-EM model provides a molecular view to explain the 100-fold lower $K_D$ of OdhI on OdhA[41] *vs.* GarA on *Ms*KGD, as determined by surface plasmon resonance[32].

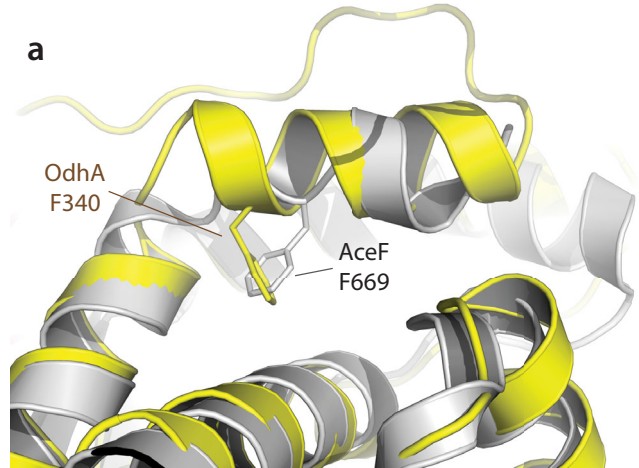

**a**

OdhA
F340

AceF
F669

**b**

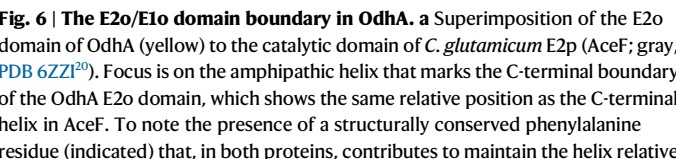

F340

**Fig. 6 | The E2o/E1o domain boundary in OdhA. a** Superimposition of the E2o domain of OdhA (yellow) to the catalytic domain of *C. glutamicum* E2p (AceF; gray, PDB 6ZZI[20]). Focus is on the amphipathic helix that marks the C-terminal boundary of the OdhA E2o domain, which shows the same relative position as the C-terminal helix in AceF. To note the presence of a structurally conserved phenylalanine residue (indicated) that, in both proteins, contributes to maintain the helix relative

orientation through intramolecular interactions. Such orientation was shown to be key to the loss of high-order oligomerisation in actinobacterial E2p enzymes. **b** Sequence logo derived from a multiple sequence alignment of OdhA orthologues from representative members of the Actinobacteria phylum (see Supplementary Fig. 17). The logo is here limited to the OdhA residues surrounding the conserved Phe340.

## Interactions between the two catalytic centers

The homohexameric arrangement of OdhA challenges current paradigms about the composition and protein-protein interactions within PDH and ODH complexes and raises questions regarding the coordination of the different catalytic activities carried out by the same polypeptide. We previously reported how the E2o domain contributes to regulate E1o activity in *Ms*KGD by restraining protein motions involved in the transition from the resting to the active state[28]. An arginine residue (Arg781) situated on the loop preceding the αE helix was indeed shown to mediate contacts with the E2 domain, and the analysis of available sequences of OdhA homologs from Actinobacteria shows the conservation of this residue (Supplementary Fig. 17). In the resting state OdhA$_{\Delta 97}$ crystal structure, Arg781 hydrogen bonds to the main chain oxygens of Arg151 and Thr152, but these interactions are not observed in the EM models (including the OdhA-OdhI complex), where the distance of the guanidium group to the Arg151 carbonyl oxygen is around 7 Å. A salt bridge in between Asp777 (helix αE) and Arg153 from the E2o domain is observed instead (Supplementary Fig. 19). Intrigued by these differences that suggest interdomain mobility, we performed 3D variability analysis[43] on all our EM datasets. The results indicate indeed twisting of the E1o domains around the hexamer plane, as well as tilting movements of the longitudinal axes of the same E1o domains, which deviate from their average position on the three-fold axis of the hexamer (Supplementary Movie 3), reinforcing the hypothesis that interdomain flexibility is a major contributor to protein dynamics, which, by remodeling the contact network, may contribute to enzyme regulation.

The homohexameric OdhA architecture also provides a further example of conservation of structural motifs at the domain interfaces. We showed recently how actinobacterial E2p enzymes lose their typical high molecular weight oligomerization due to a specific 3-residue insertion at their C-terminus, and as a consequence they are reduced to their minimal catalytic homotrimeric state[30]. Specifically, the insertion makes the terminal 3$_{10}$ helix, involved in symmetric trimer-trimer interactions, to deviate from its position making intramolecular contacts instead. It is worth noting that a similar situation is observed at the E2o-E1o interface in OdhA, where the C-terminal amphipathic α-helix from the E2o domain adopts a conformation substantially equivalent to the one observed in AceF (Fig. 6a), its internal face being held against helix α3 from the same domain (OdhA residues Phe159-

Ala173), mostly by hydrophobic interactions. Moreover, a phenylalanine residue (Phe340 in OdhA) occupies a structurally equivalent position to Phe669 in AceF (Fig. 6a), shown to be a key conserved residue of the 3-amino acid insertion in actinobacterial E2. Likewise, as observed in AceF, the Phe340 OdhA side chain contributes directly to fill the hydrophobic pocket which, in canonical E2 enzymes, would accommodate the incoming C-terminal 3$_{10}$ helix from the facing trimer, in the so-called 'knobs and socket' interaction. Consistently, sequence alignment of OdhA orthologues shows the conservation of the phenylalanine residue (Fig. 6b; Supplementary Fig. 17), in agreement with previous considerations[30] and thus confirming the role of the phenylalanine-containing insertion (PCI) as a structural motif in actinobacterial E2 enzymes.

## Discussion

2-oxoacid dehydrogenase complexes have long made a textbook example of megadalton-sized, universally conserved multienzymatic machineries, each dedicated to specific, yet conserved, three-step reactions located at the core of central metabolism. These complexes have, so far, been thought to be centered around large hollow cores composed of multiple copies of specific E2 (acyltransferase) enzymes, which, through long and flexible linkers that bear lipoyl and interaction domains, anchor the E1 (ThDP-dependent decarboxylases) and E3 (lipoamide dehydrogenase) components on the outside surface. Despite the first evidence of such architecture dates back to pioneering investigations in the sixties[7,8], the inherent flexibility of the E2 interdomain linkers and the transient nature of some of the protein-protein interactions has long hampered a detailed understanding of such complexes. Even their stoichiometry, including in well-studied model organisms like *E. coli*, has long been a matter of debate. Thanks to methodological advances in single particle EM, and to the application of integrative approaches, the last couple of years have however seen significant advances in the field, ranging from the composition of eukaryotic E2p and E2o cores[22–24,44–46], to snapshots of protein interactions and dihydrolipoyl-lysine entering the E2 active site[2,21,24,47]. None of the aforementioned studies, however, dealt with complexes deviating significantly from the well-established general architecture of oxoacid dehydrogenases. Here, by a combination of high-resolution X-ray crystallography and single particle EM analysis, we show how evolution, through shuffling and fusion of domains, combined fully

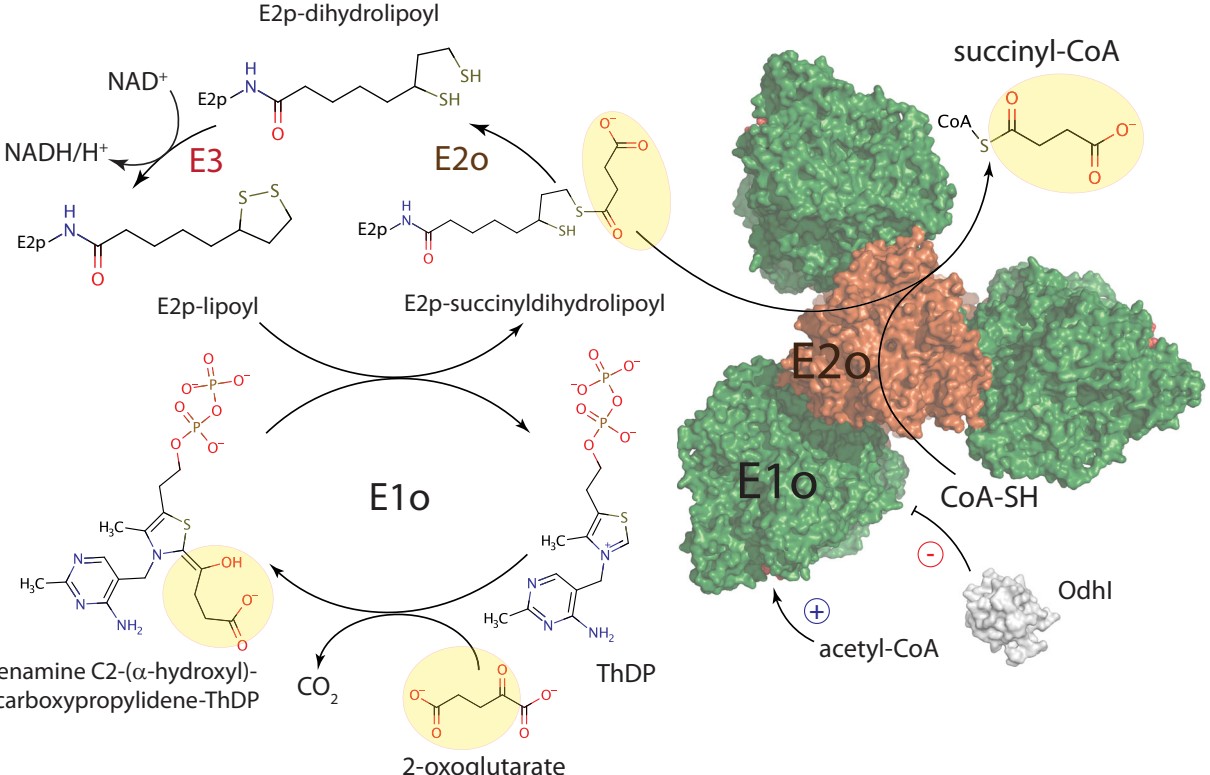

**Fig. 7 | Overall scheme of the 2-oxoglurate dehydrogenase (ODH) reaction in Actinobacteria.** Schematic illustration of the reaction steps involved in the oxidative decarboxylation of 2-oxoglutarate with generation of succinyl-CoA and NADH/H⁺ (ODH reaction), with emphasis on the catalytic steps catalyzed by OdhA. These include the ThDP-dependent decarboxylation of the 2-oxoglutarate substrate followed the reductive acylation of the lipoyl group provided by E2p (both catalyzed by the OdhA E1o domain), and the transfer of the succinyl moiety to the CoASH acceptor (catalyzed by the OdhA E2o domain). The dihydrolipoyl group is then re-oxidized by the flavoenzyme E3 (Lpd) with the generation of NADH/H⁺ from NAD⁺. On the right, the OdhA-CoA single particle cryo EM model is shown as atomic surface, with different colors per domain (green for E1o, orange for E2o).

functional E2o and E1o activities in a single polypeptide, generating a 'two-in-one' enzyme that only depends on oxidized lipoyl groups provided by E2p and regenerated by the E3 component (Fig. 7). Despite the presence of such domain fusion being first reported in the nineties[25], the three-dimensional organization of such a 0.8 MDa oligomeric enzyme had so far resisted attempts at structural characterization. We also provide atomic-resolution details of bound cofactors, substrate analogs and products, and provide insights into the allosteric regulation mechanism driven by an FHA module, another distinct feature of Actinobacteria. The unique, homohexameric three-blade propeller shaped state of KGD and OdhA not only stands out for its size and symmetry, but also raises new questions about the evolution of PDH and ODH complexes. The coexistence of ThDP-dependent dehydrogenase and succinyltransferase domains on the same polypeptide (Fig. 7), and their interactions, has obvious implications not only in terms of substrate channeling and catalytic efficiency, but also in terms of regulation, as first suggested by our previous studies on mycobacterial KGD[28]. Furthermore, here we show how the succinyltransferase domain makes use of the same C-terminal structural motif we previously identified in corynebacterial AceF, and adds to the hypothesis of a link between the presence of an OdhA-like, 'two-in-one' enzyme and a mixed PDH/ODH supercomplex, whose presence, initially proposed on the basis of copurification experiments in *C. glutamicum*[27], has been supported by increasing experimental evidence[26,28,35]. It is therefore tempting to speculate that the correlation between the presence of an OdhA-like enzyme bearing both E2o and E1o activities, and a reduced E2p core may be related to the size and hexameric architecture of OdhA, possibly incompatible with its interaction with a canonical cubic or dodecahedral PDH core. How the

same E2p lipoyl domains may be able to serve the catalytic sites of OdhA as well as those of E1p and E2p itself is indeed one of the most interesting open questions[48]. The reasons usually evoked as the major advantages brought in by the assembly of large, multimeric complexes, i.e. active site coupling and efficient substrate channelling, may actually turn out to be, as pointed out following the publication of our previous study, just 'one side of the coin'[48]. The physical proximity of the PDH and ODH centers may facilitate the coregulation of the pyruvate and oxoglutarate nodes, as indicated by the positive regulation of both OdhA and *Ms*KGD by acetyl-CoA, which suggests the presence of positive feedback mechanisms. The next challenging goals will include determining how OdhA and the other components of the PDH/ODH supercomplex may be physically and temporally assembled in a supramolecular structure, and whether such an assembly could interact with other cellular structures. Exploring these new avenues will lead to a better understanding of fundamental biological processes like the regulation of central metabolism, as well as to novel therapeutic approaches that may target Actinobacteria-specific protein-protein interactions.

## Methods
### Plasmid construction
Expression constructs pET-28a-TEV/OdhA and pET-28a-TEV/OdhI were generated by Genscript (Leiden, the Netherlands), providing a sequence coding for the TEV protease cleavage site (ENLYFQG) between the vector encoded His₆-tag and the N-terminus of either *C. glutamicum* ATCC13032 *odhA* (Uniprot accession no. Q8NRC3, residues 1-1221) or *odhI* (Uniprot accession no. Q8NQJ3, residues 1-143). The pET-28a-TEV/OdhA_Δ97 construct (coding for OdhA residues

98-1221) was also generated by Genscript from pET-28a-TEV/OdhA. The *C. glutamicum* ATCC13032 *aceE* (Uniprot accession no. Q8NNF6, residues 1-922) and *lpd* (Uniprot accession no. Q8NTE1, residues 1-469) open reading frames were amplified by PCR (Supplementary Table 2) and inserted, by restriction-free cloning[49], into the pET-32a derived pT7 expression vector providing a TEV cleavage site at the 5' end of the target gene[50]. Constructs were verified by DNA sequencing.

## Protein purification

Full-length *Ms*KGD was overexpressed in *E. coli* BL21(DE3)pLysS and purified as previously described[28]. Both OdhA expression constructs (pET-28a-TEV/OdhA and pET-28a-TEV/OdhA$_{\Delta 97}$) were introduced into *E. coli* BL21(DE3), and protein expression achieved following the same autoinduction scheme[51]. Recombinant proteins were also purified following the same protocol. After an overnight incubation at 30 °C in 2YT-based autoinduction medium containing 50 µg/ml kanamycin, cells were harvested and frozen at −80 °C. Cell pellets were resuspended in 50 ml lysis buffer (25 mM Tris pH 8.5, 300 mM NaCl, 25 mM imidazole, supplemented with benzonase and EDTA-free protease inhibitor cocktails (Roche)) at 4 °C, and lysed by a CF2 cell disruptor (Constant Systems Ltd.). The lysate was centrifuged for one hour at 13,000 ×g at 4 °C. The clear supernatant was loaded onto a Ni-NTA affinity chromatography column (1 ml HisTrap FF crude, Cytiva), and his-tagged proteins were eluted with a linear gradient of buffer B (25 mM Tris pH 8.5, 300 mM NaCl, 400 mM imidazole). The eluted fractions containing the protein of interest were pooled and TEV protease, produced as described[52], was added at 1:30 w/w ratio. The sample was then dialyzed overnight at 4 °C against 20 mM Hepes pH 7.5, 500 mM NaCl, 1 mM DTT using 'SnakeSkin' dialysis tubing with a 7 kDa molecular weight cut-off (ThermoFisher). His$_6$-tagged cleavage products as well as TEV protease were removed with Ni-NTA agarose resin (Qiagen) on gravity flow disposable plastic columns. The sample was then concentrated and loaded onto a Sephacryl S-400 HR 16/60 size exclusion (SEC) column (Cytiva) pre-equilibrated in 20 mM Hepes pH 7.5, 500 mM NaCl (20 mM Hepes pH 7.5, 300 mM NaCl for OdhA$_{\Delta 97}$). Fractions corresponding to the OdhA peak were checked on SDS-PAGE (Supplementary Fig. 20), pooled and concentrated. The resulting sample was either used directly for cryo-EM grid preparation, or flash-frozen in small aliquots in liquid nitrogen and stored at −80 °C.

OdhI was overexpressed by autoinduction in *E. coli* BL21(DE3) grown in the same 2YT-based medium as OdhA, but overnight culture at 14 °C. The purification also followed the protocol above, but size-exclusion chromatography was performed on a HiLoad Superdex 75 16/60 column run in 25 mM Tris-HCl pH 8.5, 150 mM NaCl. Likewise, Lpd (E3) was also overexpressed by autoinduction in 2YT-baed medium supplemented with 50 µg/ml carbenicillin, with overnight culture at 18 °C; the protein was purified following the same steps, except for size-exclusion chromatography performed on a HiLoad Superdex 200 16/60 column run in 50 mM Tris-HCl pH 8.5, 150 mM NaCl, 5% glycerol. AceE (E1p) was overexpressed, in *E. coli* BL21(DE3) grown in LB medium containing 50 µg/ml carbenicillin, by the addition of 0.5 mM IPTG at the optical density of ~0.6 (600 nm), followed by 18 h growth at 18 °C. The recombinant protein was purified following the same protocol as OdhA, with the size-exclusion chromatography step performed on Sephacryl S-400 HR 16/60 equilibrated in 50 mM Tris-HCl pH 8.5, 150 mM NaCl, 5% glycerol. *C. glutamicum* full-length AceF (E2p) and *M. smegmatis* GarA were expressed and purified as previously described[30,32].

## Oxidative decarboxylation and 2-oxoacid dehydrogenase assays

Oxidative decarboxylation activity of OdhA was determined by measuring 2,6-dichlorophenolindophenol (DCPIP) reduction at 600 nm and 25 °C[26]. The reaction medium contained 0.1 M KH$_2$PO$_4$ pH 7.0, 1 mM ThDP, 1 mM MgCl$_2$, 0.25 mM DCPIP, and 1 mM 2-oxoglutarate.

The reaction was started by addition of reaction medium to a well containing 9 µg of OdhA to the final volume 200 µl. Blank reaction rate was measured in the reaction medium omitting 2-oxo acid. The extinction coefficient of DCPIP used for calculations is 20.6 mM$^{-1}$cm$^{-1}$. 2-oxoacid dehydrogenase activity (PDH or ODH) was determined in conditions adapted from previous reports[26,27]. The assay medium contained 50 mM TES buffer pH 7.7, 10 mM MgCl$_2$, 3 mM L-cysteine, 0.9 mM TPP, 50 µM chlorpromazine, 2 mM NAD$^+$, 0.2 mM coenzyme A, 10% glycerol and either 1.5 mM pyruvate or 1.5 mM 2-oxoglutarate to measure PDH or ODH activity, respectively. To achieve an approximately equimolar ratio while accounting for the presumed oligomeric state of each enzyme, 414.8 µg of OdhA, 105.6 µg of AceE, 109.8 µg of AceF, and 52.2 µg of Lpd were mixed at final concentration 11.2 mg/ml. Before activity measurements, the mixture was incubated on ice for at least 30 min. Reactions were started by adding the reaction medium to the protein mixture (10–40 µl) in a final volume of 200 µl, and were followed by NADH absorbance at 340 nm at 30 °C, using an Infinite M1000Pro reader (Tecan). Blank reaction rate was measured in the reaction medium omitting the 2-oxoacid substrate. Extinction coefficient of NADH used for calculations is 6.22 mM$^{-1}$cm$^{-1}$.

## Crystallization

Initial crystallization screenings were performed at 18 °C by vapor diffusion in 96-well plates, according to established protocols at the Crystallography Core Facility of the Institut Pasteur[53]. Crystals of the OdhA$_{\Delta 97}$-CoASH complex were obtained from a 26 mg/ml OdhA$_{\Delta 97}$ solution, supplemented with 5 mM CoASH and crystallized in 0.1 M Hepes-NaOH pH 7.5, 5% (w/v) PEG 4000, 30% (v/v) methylpentanediol (MPD) by the sitting drop method; for the *Ms*KGD-GarA complex, crystals were obtained from a 10 mg/ml *Ms*KGD solution supplemented with 2 mM ThDP, 5 mM MgCl$_2$ and *M. smegmatis* GarA (1:1 molar ratio), and crystallized, through the hanging drop method, in 0.1 M bicine pH 8.0, 30% (v/v) PEG550MME, 0.2 M NaCl.

## X-ray diffraction data collection and structure solution

Diffraction datasets were acquired either on the beamline ID30A-3 at the ESRF synchrotron (Grenoble, France), or on the beamline Proxima-1 at the SOLEIL synchrotron (Saint-Aubin, France). Data integration and scaling were performed with *autoPROC*[54], applying anisotropic scaling via *STARANISO*. Structures were solved by molecular replacement through the program *PHASER*[55], using the previously released coordinates of the *Ms*KGD$_{\Delta 115}$ homodimer (PDB 2XT6[28]) as the search model for both datasets (OdhA$_{\Delta 97}$-CoA and *Ms*KGD:GarA). *M. smegmatis* GarA coordinates were retrieved from the previously published *Ms*KGD$_{\Delta 360}$:GarA complex (pdb 6I2Q[32]). Manual model rebuilding and ligand placement in electron density maps was entirely performed with *COOT*[56]. Refinement was carried out with *BUSTER*, applying local structure similarity restraints for non-crystallography symmetry (NCS)[57] where appropriate, and a Translation-Libration-Screw (TLS) model. Chemical dictionaries for ligands were generated with the Grade server (http://grade.globalphasing.org). Validation of models was performed with *MOLPROBITY*[58] and the validation tools in *PHENIX*[59]. Data collection, refinement and model statistics are indicated in Table 1. Software was distributed by the SBGrid consortium[60].

## Negative staining EM

5 µl of purified OdhA sample, at concentrations of either 0.05 mg/ml or 0.01 mg/ml, were applied over 400-mesh copper carbon coated grids (Electron Microscopy Sciences) that were previously glow discharged at 2 mA for 20 s. Grids were washed twice in 10 µl water for 40 s, then stained in a 2% uranyl acetate solution (twice for 40 s). Grids were then blotted using a Whatman 1 filter paper and air dried for 5 min. Micrographs were acquired on a Tecnai T12 transmission electron microscope (ThermoScientific), operating at 120 kV, at magnification rates comprised between 30,000× and 180,000×.

## Cryo-EM sample preparation and data collection

OdhA samples were vitrified at a concentration of 12.0 mg/ml (protein without ligands, incubated with CoASH or succinyl-CoA), or 8.0 mg/ml for the OdhA-SP complex. The OdhA-OdhI complex was prepared by incubating a mixture of the two proteins at molar ratio 1:10, which was then subjected to size-exclusion chromatography on a Superose 6 increase 5/150 GL column (Cityva), run in 20 mM Hepes pH 7.5, 500 mM NaCl. UltrAuFoil R1.2/1.3 300 mesh gold grids (Quantifoil) were used for OdhA alone, OdhA-CoASH or OdhA-succinyl-CoA, while OdhA-SP and OdhA-OdhI were vitrified on Lacey carbon 200 mesh grids (Electron Microscopy Sciences). Grids were glow discharged for 25 s at 50 W (UltrAuFoil R1.2/1.3) or 10 s at 5 W (Lacey) under 35.0 sccm Ar, with a Solarus II plasma cleaner (Gatan). Vitrification was carried out using a Vitrobot Mark IV (ThermoScientific), applying 3 μl of protein sample to the grid surface at a temperature of 4 °C and humidity level of 100%. Grids were then blotted (during 4 s at blot force 0 for Lacey grids, blot force 2 for UltrAuFoil R1.2/1.3 grids) and plunge-frozen into liquid ethane. Data from all samples but OdhA-succinyl-CoA were collected at the Nanoimaging Core facility in Institut Pasteur on a Titan Krios electron microscope (ThermoScientific), operated at 300 kV and equipped with a K3 direct electron detector (Gatan) operating in counted mode. The OdhA-succinyl-CoA dataset was instead collected on a Titan Krios microscope located at the ESRF (Grenoble, France), also running at 300 kV and equipped by a K3 detector operating in the counted super-resolution mode[61]. The software EPU (ThermoScientific) was used to pilot data collection in all cases. A summary of data collection and model refinement parameters is reported in Table 2.

## Single particle analysis of cryo-EM data

All single particle cryo-EM datasets were processed through cryoS-PARC version 3.2[62]. Motion correction was performed using full-frame motion correction and CTF estimation were performed using patch CTF estimation. Using the curate exposure feature, 12666 out of 13348 for OdhA alone, 7996 out of 12202 for OdhA-CoASH, 8796 out of 11827 for OdhA-succinyl-CoA, 15025 out of 16647 for OdhA-SP, and 14842 out of 19443 for OdhA-OdhI complex were selected for further analysis. A first round of 'blob particle picking' was performed, and after 2D classification, the most populated classes were selected for template-based particle picking against a dataset containing the selected micrographs. Particles were extracted applying a box size of 384 Å, except for the OdhA-SP sample for which the box size was set at 448 Å. The extracted particles were cleaned using the 'inspect pick' function of Cryosparc and several rounds of 2D classification with selection of higher resolution classes were applied, selecting classes corresponding to a complete, hexamer shape of OdhA in different orientations. The selected particles were used, after local motion correction, to build an ab-initio model and non-uniform 3D refinement applying D3 symmetry, while optimizing per-particle defocus and per-group CTF parameters. For all datasets, local refinement was performed with a soft mask covering the OdhA E1o domain dimer. A composite map including the three OdhA E1o dimers was generated by the 'combine focused maps' tool in PHENIX (v. 1.20-4459-000)[59].

## Single particle EM model building and refinement

OdhA coordinates obtained from X-ray crystallography were fitted into the corresponding cryo-EM density maps using UCSF Chimera (v1.13.1)[63]. For the OdhA-OdhI complex, *C. glutamicum* OdhI coordinates were retrieved from the available crystal structure (pdb 4QCJ[64]). Following a first round of rigid-body fitting of the E2o and E1o OdhA domains, and OdhI were appropriate, the models were improved by iterative rounds of restrained real-space refinement in *PHENIX*, and alternating rounds of model building with manual adjustment in *COOT* and further model refinement through the Servalcat pipeline[65] in the CCP-EM suite[66]. Model validations were performed using the specific tools in the *PHENIX* and CCP-EM suites. Figures were prepared using UCSF ChimeraX v.1.3[67], and PyMOL v. 2.5.4[68], distributed by the SBGrid consortium[60].

## Surface plasmon resonance binding assay

Experiments were performed using a Biacore T200 instrument (Cytiva) and NTA sensorchips equilibrated at 25 °C in OdhA storage buffer (20 mM Hepes pH 7.5, 500 mM NaCl), complemented with 100 μM EDTA and 0.2 mg/ml BSA. Two flow cells of the chip were first activated by running a 1 mM $NiCl_2$ solution for 2 min at 5 μl/min, and loaded with OdhA-His$_6$ (200 μg/ml) for 10 min at 5 μl/min, reaching densities of 8800–10,000 resonance units (RU, ≈pg/mm$^2$). 5 concentrations of SP were then injected sequentially in single cycle kinetics mode at 30 μl/min for 2 min each, followed by a 2 min buffer wash to monitor the dissociation of the OdhA-SP transient complex. The sensorchip was finally fully regenerated by injecting 0.5 M EDTA, 0.1% SDS twice for 2 min at 5 μl/min, allowing it to be reused for a new experimental cycle. Sensorgrams were processed using the BiaEvaluation software. The concentration-dependence of steady-state SPR signals (Req) was analyzed using the following equation: Req = Rmax * C/($K_D$ + C), where C is the SP concentration and Rmax the fitted maximal SPR signal at infinite SP concentration.

## Analytical ultracentrifugation

Sedimentation velocity (SV) analytical ultracentrifugation assays were performed using a Beckman Coulter ProteomeLab XL-I analytical ultracentrifuge equipped with UV-Vis absorbance and Raleigh interference detection systems, using an 8-hole Beckman An-50 Ti rotor at 20 °C. Experiments were performed at 30,000 rpm. Seven concentrations (from 4 mg/ml to 0.0625 mg/ml, serial two-fold dilutions) were prepared for this experiment in the OdhA buffer (20 mM Hepes pH 7.5, 500 mM NaCl) and loaded into analytical ultracentrifugation cells. During the run, SV was followed by measuring absorbance at 280 nm for sample with concentration from 4 mg/ml to 0.25 mg/ml and at 225 nm for sample with concentration at 0.125 mg/ml and 0.0625 mg/ml. The software SEDFIT v. 15.01[69] was used to calculate the sedimentation coefficient distribution C(s), then corrected to standard conditions to get the final standard values. Coefficients were plotted as a function of the different concentrations and an extrapolation to zero concentration was made to obtain the standard value for the main oligomer. From these values, molecular mass and friction ratio were obtained.

## Protein sequence analyses

Sequence analyses were carried out on a database representing all Actinobacteria diversity present at the National Center for Biotechnology (NCBI) as of February 2021, containing 133 taxa (five species per class). To identify OdhA homologs, the jackhmmer tool from the HMMER package (v3.3.2)[70] was employed, using the GenBank[71] sequence BAB98522.1 as the query. The hits were aligned with mafft (v7.475)[72] accurate option L-INS-I. The MSA was manually curated, removing sequences that did not align. A sequence logo of OdhA was created based on the MSA through the online tool WebLogo3[73]. The secondary structure of protein OdhA was mapped on the MSA using the online tool ESPript[74].

## Reporting summary

Further information on research design is available in the Nature Portfolio Reporting Summary linked to this article.

# Data availability

Atomic models described in this study, accompanied by the corresponding structure factors (for X-ray crystallographic structures) and maps (for single particle cryo-EM) have been deposited to the Protein Data Bank (PDB) / Electron Microscopy Data Bank (EMDB), under the

following accession codes: *Ms*KGD-GarA (crystal structure), PDB 8P5R; OdhA$_{\Delta97}$ (crystal structure), PDB 8P5S; OdhA (cryo-EM structure with no added ligands), PDB 8P5T / EMD-17452; OdhA-CoASH, PDB 8P5U / EMD-17453; OdhA-succinyl-CoA, PDB 8P5V / EMD-17454 (raw data available at [https://doi.org/10.15151/ESRF-ES-514136397]); OdhA-SP, PDB 8P5W / EMD-17455; OdhA-OdhI complex, PDB 8P5X / EMD-17456. Previously published structural models cited in the paper include the PDB entries 1EAB, 2XTA, 2XT6, 4QCJ, 6I2Q, 6R29, 6ZZI. Source data are provided as a source data file, containing raw data relative to enzymatic activity measurements, analytical ultracentrifugation (Supplementary Fig. 2), SPR experiments (Supplementary Fig. 12b/c) and OdhA size-exclusion chromatography profile (Supplementary Fig. 20) are provided as a Source Data file with this paper.). Source data are provided with this paper.

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

## Acknowledgements

This work was funded by the ANR projects SUPERCPLX (ANR-13-JSV8-0003) and METACTINO (ANR-18-CE92-0003), both granted to M.B., and by institutional funds from the Institut Pasteur, the CNRS and the Université Paris Cité. We are grateful to the core facilities at the Institut Pasteur C2RT (Center for Technological Resources and Research) and their respective staff, namely to A. Haouz, P. Weber and C. Pissis for performing crystallization screenings, B. Raynal, S. Brûlé and S. Hoos for their assistance in analytical ultracentrifugation and buffer optimization, P. England for his help with surface plasmon resonance experiments, J.-M. Winter, S. Tachon and M. Vos for assistance with EM data collection. We are also grateful to N. Barilone, I. Miras and P. Vilela for their initial work in construct generation and protein expression, to A. Bezault, G. Péhau-Arnaudet and C. Rapisarda for their help in grid preparation and EM data collection, and to M. Bott and B.J. Eikmanns for insightful discussions. We acknowledge the synchrotron sources Soleil (Saint-Aubin, France), and ESRF (Grenoble, France) for granting access to their facilities, and their respective staff for helpful assistance, in particular E. Kandiah for performing EM data collection on CM01 (ESRF). The NanoImaging Core at Institut Pasteur was created with the help of a grant from the French Government's 'Investissements d'Avenir' program (EQUIPEX CACSICE, ANR-11-EQPX-0008), and is acknowledged for support with cryo-EM sample preparation, image acquisition and analysis. L.Y. and A.B. have both been affiliated to the Pasteur – Paris University (PPU) International PhD program; L.Y. was funded by the Wuhan Institute of Biological Products Co. Ltd. (Wuhan, People's Republic of China), subsidiary company of China National Biotec Group Company Limited, and by a doctoral fellowship from the China Scholarship Council (CSC). Succinyl phosphonate was a kind gift from V. Bunik (Lomonosov University, Moscow); TEV expression plasmid was provided by H. Berglund (Karolinska Institute, Stockholm).

## Author contributions

L.Y., T.W., A.B and E.M.B. produced and purified recombinant proteins; L.Y., T.W., A.B., F.G. and M.B. collected data; L.Y. prepared cryo-EM grids; L.Y. and A.M. analyzed cryo-EM data; L.Y., T.W., A.M. and M.B. refined models; L.Y. performed biophysical experiments; A.B. performed enzymatic activity assays; D.M. performed bioinformatic analysis; P.M.A. provided advice on data interpretation and contributed to the manuscript; M.B. supervised the work and wrote the paper. All authors reviewed the manuscript and agreed on its content.

## Competing interests

The authors declare no competing interests.
