## [Peer Review File · Nature Communications]

High resolution cryo-EM and crystallographic snapshots of the actinobacterial two-in-one 2-oxoglutarate dehydrogenaseREVIEWER COMMENTS

Reviewer #1 (Remarks to the Author):

This work is a very significant contribution to the field, according to this reviewer, who is active on this very field. The work supports the conclusions and claims. This reviewer can see no apparent flaws in the data analysis, interpretation and conclusions. The methodology is very sound and in fact state-of-the-art. The work absolutely meets the expected standards of the respective field and the Journal. There are enough details provided in the methods for the work to be reproduced. English language is flawless.

Points to be addressed:

- line 44: it would be unfortunate to mention "hydrophobic amino acids" here, I would mention the three actual amino acids involved.**
- In this same sentence, I would also mention the 2-oxoadipate dehydrogenase complex and relevant recent references.**
- Regarding naming convention: I would mention, at least in the Introduction, that these 2-oxo acid dehydrogenases also run in the literature under the name of alpha-keto acid dehydrogenases.**
- It is still not completely clear to this reviewer from the manuscript: The OdhA E2o domain has no lipoic acid binding stretch(es) [N-terminal sequences homologous to typical lipoyl domains] at all according to multiple sequence alignment? Has anybody ever tried a lipoyl ligase reaction on this protein? Mass spectrometry cannot really reveal any lipoylated site on this protein from natural sources?**
- "worth to note" should rather be "worth noting", at many places in the manuscript.**
- line 327: the recent hE2o cryo-EM structure (Nagy B et al., BBA, 2021) should probably be cited here, too, as was also cited in the Suppl. material.**
- line 329: here I would also cite Nagy et al., BBA, 2021 (again) as well as Nemeria NS et al., Int. J. Mol. Sci, 2022, as very relevant references to the subject.**

Reviewer #2 (Remarks to the Author):

Review of manuscript entitled: "High resolution cryo-EM and crystallographic snapshots of the large actinobacterial 2-oxoglutarate dehydrogenase: an all-in-one fusion with unique properties", by Yang et al.

In this work, the authors have structurally characterized in near-atomic resolution, the actinobacterial 2-oxoglutarate dehydrogenase complex. To this purpose, they have employed both x-ray crystallography and single-particle cryo-EM reconstructions to derive complementary insights into the complex's 3D organization, side-chain conformations, substrate placement and domain flexibility, revealing intriguing and novel information about the complex's structure.

The manuscript is very well written and extremely detailed in its presentation of all the different structural insights that have been derived during this work. Additionally, the authors have provided full validation reports for all of their reconstructions, something that even though should be mandatory, it is not, but is very appreciated. The reports further corroborate the very high quality of their presented data.

This reviewer is a bit intrigued by the choice of journal, as the manuscript feels like a better fit for a journal that is more targeted towards the structural biology community, as, for example, Nature Structural and Molecular Biology. The high-level and detailed structural descriptions in the text will probably confuse a reader who is not very familiar with structural biology. Despite that, the novelty contained in this paper makes it suitable for publication in Nature Communications.

With all this said, there are some comments that this reviewer believes should be

answered, in order to further improve the quality of the manuscript overall:

Major comments:

1) In the crystal structure, the authors report an alternate conformation of the KGD E1o domain N-ter segment, observed to form a short helical hairpin. They suggest that this can participate in intermolecular interactions and cite a work performed for the human E1o. In the eukaryotic E1o, the N-ter segment is disordered and is possibly contributing to the tethering of the E1o dimer to the periphery of the E2o cubic core (see recent publication by Hevler et al. 2023, Open Biology [10.1098/rsob.220363]) as the E2o N-ter lipoyl- arm lacks a peripheral subunit binding domain (in contrast to the presence of PSBD in PDHc E2). In the case of the actinobacterial E1o, as shown in Fig.1 of the manuscript, this N-ter helical hairpin is neither in the vicinity of E2o, or such a function, similar to the eukaryotic counterpart, should be required as the 2 proteins are in essence fused. Could the authors better describe how this region could participate in in-complex intermolecular interactions? Are there other proteins reported to participate in the formation of the oxoglutarate dehydrogenase complex in actinobacteria?

2) Even though the authors have captured in their structures bound substrates in the active sites of the complex, they have not performed any additional activity assays. In this reviewer's opinion, it is critical to show that the presented complex is active, and to this purpose activity assays for the complete oxoglutarate dehydrogenase reaction. In the text (line 112-113) they mentioned activity results for the truncated, E1 domain that they have previously published. Again, in lines 181-182 they cite they previous work on E1o for the same purpose, to define the early/late stages of E1o conformations, which they again observe here.

Similar work should be performed with the complete OdhA protein to verify that the presented assembly is truly an active state and they do not only observe isolated conformations imposed only by E1o activity. This is also in connection to the previous comment (1), as no proof is given in the manuscript that OdhA is the sole component required for the production of succinyl-CoA. Showing that the complex is truly active and can perform the complete reaction would be critical also as verification that this assembly is a true native state.

3) The authors performed 3DVA in order to observe the domain movements within the complex and then discuss the implications of this movement mostly in regards to the inhibition mechanism of OdhA. In regards to the "Interactions between the two catalytic centres" section, where again 3DVA is employed to describe coordination between the catalytic activities, the authors extensively describe the observed contact network but at the end of the section (lines 294-295) only briefly mention that this this type of domain movement-derived contact remodeling would be critical for the complex's regulation. The authors circle the subject, but do not provide at least a hypothesis (based on their observations) about reaction intermediate channeling, given an apparent lack of a lipoyl- arm that will move reaction intermediates from the E1o domain to the E2o. Do they believe that, due to the close distance of the respective active sites, substrates only diffuse? Does the full N-ter OdhA segment (~100 residues) contain a sequence similar to the eukaryotic lipoyl- domain? Have they tried to model it with AlphaFold2? Could they elaborate on the topic?

4) Are the authors certain that in lower thresholds there is no density for the OdhA N-ter? Additionally, have they observed asymmetric densities in C1 reconstructions of the complex? Given enough particle abundance in their dataset, a C1 reconstruction may provide insights into asymmetric features of the complex, which, if existing, this reviewer feels they are worth reporting.

5) Have the authors tried to also align the GarA with the lipoyl- domain of eukaryotic E2o? Their overall folds appear quite similar.

6) In the discussion (lines 361-362) the authors speculate on the assembly's capability of interaction with other cellular structures. Can they suggest what those may be? Are there mentions in the literature concerning the specific topic?

Other comments:

7) The figures, even though quite beautiful and of very high resolution, are often confusing. Additionally, maybe some more panels should be added to increase readability, so that the reader can more closely follow the detailed structural descriptions of the text. In more detail:

a. Suppl. Fig. 1: GarA domains should be colored differently for the MsKGD-GarA (orange) complex for better visibility.

b. Fig. 1: There should be an additional panel with the N-ter helical segment zoomed in. At this state it is quite hard to observe it well.

c. Suppl. Fig. 4: Add a sub-panel showing the map density covering the β -sheet.

d. Fig. 2: Sub-panels should be marked with A, B, C, D and cited appropriately in the text.

e. Fig. 2: Show an additional sub-panel with more detail into the contacts of the short α -helix (lines 134-135).

f. Add (probably in sup. Fig. 9) a panel with side-chain density coverage examples to demonstrate visually lines 168-171.

g. Suppl. Fig. 16: Use different color, or better use a box to highlight conserved residues below the alignment, they get lost among the other red-colored features.

h. Fig. 5a (line 228): Only shows the map/model generated in this paper, no comparison. It would be useful to add it. Otherwise, Sup. Fig. 1 should be cited here (it shows a general alignment).

i. Fig. 5b,c: Authors should consider splitting these figures and lower the amount of simultaneous side-chains displayed in each panel, in accordance with the text. The figure panel is quite small and it is hard to easily locate each side-chain discussed in-text.

j. Materials and Methods (lines 422-423): Authors should add a sup. fig. with SDS gels and elution profile annotating fractions.

8) Lines 117-119 (OdhA presenting... OdhA model): Should probably be rewritten to start with: "As OdhA indeed presents the same.....".

9) Line 164: (fig 3c) in bold.

10) Line 173: (fig. 1b) in bold.

11) Line 339: stands out, not off.

12) Line 347: whose, not which.

13) Throughout the text: some words have probably been auto-corrected during writing of the manuscript to their French spelling. Inspect the text again for such occurrences.

14) Line 531: state rotor speed in xg, not RPM.

15) Line 545: L-INS-I, not linsi.

Reviewer #3 (Remarks to the Author):

This manuscript describes direct, structural observation of 2-oxoglutarate dehydrogenasem, from an actinobacterial species. As the first structural example of such a naturally occurring, fused system version in which two component enzymes reside on a single chain, the results are novel and may be representative of this subset. In the more frequently occurring versions the multiple enzymes interact only through substrate channeling via a swinging arm mechanism. The results can be important for this class, and the experiments were appropriately carried out, described, and the results reasonably interpreted. I would however, recommend a few

changes/clarifications.

Line 29, "folds" should be "assembles", as it involves multiple chains.

Line 50, lipoyl domains also enter E3 to restore the initial conditions which enables cycling. Thus I would change "E1 and E2" to "E1, E2, and E3" active sites.

Lines 59 & 60, stoichiometries and interactions are NOT elusive for most TETHERING (psbd) interactions, as they have been determined, visually observed, and reported. Only lipoyl domain interactions with the peripheral components remain elusive.

Lines 60 & 61, this is misleading since only CORE component E2 or E3 binding protein trimer interactions, and mainly those forming the pentamers, were visible; NO channeling interactions, which MUST involve the peripheral E1 and E3 components, were EXPERIMENTALLY observed in those studies.

Line 68, I would suggest "two-in-on" rather than "all-in-one" here, in the title, and elsewhere, since activities associated with E3 (ring closure, NADH production/release) are not present, and the paper also speculates on the use of lipoyl domains from PDH in a mixed complex to enable acyl transfer. Clearly the reported complex described can not do it "all."

Does production in E. Coli affect assembly of the non-E. Coli complex, given that a related version is also present naturally? This should be mentioned.

Fig 1 caption should state the model resolution. The citing text indicates 4.6 angstroms, although electron density is not shown in the figure.

Line 142. I think "Fig 2" should be "Fig 4" or the wording changed as Fig 2 shows no electron density.

Lines 328-329, It can be argued that features enabling substrate channeling are still largely unknown since transient lipoyl domain recognition/binding has not been reconciled, especially with E1 and E3 peripheral components. The recent findings cited are confined to the cores, and so far ALL experimental methods, including cryoEM have been dismal failures regarding channeling, which MUST involve the other components. I would tone this down. There are still many remaining issues to be resolved!

Line 332 what was "cutting edge"? I didn't catch any special sub-particle optimization or new treatment as it seemed to be routine class-averaging.

Lines 356-359. How do you know PDH and ODH are in physical proximity in cells? Co-purification can be the result of assembly after cell lysis. Has a mixed complex ever been observed, and if so, what's its size and stoichiometry? This should be discussed.

Yang, L., Wagner, T., Mechaly, A.M., Boyko, A., Bruch, E.M., Megrian, D., Gubellini, F., Alzari, P.M., Bellinzoni, M.

High resolution cryo-EM and crystallographic snapshots of the large actinobacterial 2-oxoglutarate dehydrogenase: a two-in-one fusion with unique properties

REPLIES TO REVIEWER COMMENTS

Reviewer #1 (Remarks to the Author):

This work is a very significant contribution to the field, according to this reviewer, who is active on this very field. The work supports the conclusions and claims. This reviewer can see no apparent flaws in the data analysis, interpretation and conclusions. The methodology is very sound and in fact state-of-the-art. The work absolutely meets the expected standards of the respective field and the Journal. There are enough details provided in the methods for the work to be reproduced. English language is flawless.

We would like to thank the Reviewer for taking the time to evaluate our work, and for her/his comments.

Points to be addressed:

- line 44: it would be unfortunate to mention "hydrophobic amino acids" here, I would mention the three actual amino acids involved.

The actual three amino acids, *i.e.* leucine, isoleucine and valine are now cited in the text.

- In this same sentence, I would also mention the 2-oxoadipate dehydrogenase complex and relevant recent references.

For the sake of clarity and to avoid an otherwise too long sentence, mention to the 2-oxoadipate dehydrogenase complex (and the relevant references) has been added as an additional sentence that just follows the one mentioning the three main 2-oxoacid dehydrogenase complexes (PDH, ODH, BCKDH).

- Regarding naming convention: I would mention, at least in the Introduction, that these 2-oxo acid dehydrogenases also run in the literature under the name of alpha-keto acid dehydrogenases.

That is also cited now.

- It is still not completely clear to this reviewer from the manuscript: The OdhA E2o domain has no lipoic acid binding stretch(es) [N-terminal sequences homologous to typical lipoyl domains] at all according to multiple sequence alignment? Has anybody ever tried a lipoyl ligase reaction on this protein? Mass spectrometry cannot really reveal any lipoylated site on this protein from natural sources?

We thank the reviewer for raising this point. OdhA has indeed no predicted lipoic acid binding stretch, including in the N-terminal segment that precedes the E2o domain. Previous work showed that, both in *M. tuberculosis* (Tian *et al.*, 2005, *Mol. Microbiol.* 57: 859-868) and *C. glutamicum* (Hoffelder *et al.*, 2010, *J. Bacteriol.* 192, 5203-5211), E2p (named DlaT in *M. tuberculosis*, AceF in *C. glutamicum*) is the only detectable lipoylated protein in either organism. Furthermore, it has been reported that E2p is required for both the PDH and ODH reactions not only in *C. glutamicum* (Hoffelder *et al.*, 2010), but also in mycobacteria, as shown by our previous work (Wagner *et al.*, 2011, *Chem. Biol.* 18: 1011-1020).

- "worth to note" should rather be "worth noting", at many places in the manuscript.

Corrected in both occurrences in the manuscript.

*- line 327: the recent hE2o cryo-EM structure (Nagy B *et al.*, BBA, 2021) should probably be cited here, too, as was also cited in the Suppl. material.*

The citation has been added.

*- line 329: here I would also cite Nagy *et al.*, BBA, 2021 (again) as well as Nemeria NS *et al.*, Int. J. Mol. Sci, 2022, as very relevant references to the subject.*

The sentence has been slightly modified to mention integrative approaches in addition to single particle EM, and the two suggested references have been added.

Reviewer #2 (Remarks to the Author):

Review of manuscript entitled: "High resolution cryo-EM and crystallographic snapshots of the large actinobacterial

2-oxoglutarate dehydrogenase: an all-in-one fusion with unique properties”, by Yang et al.

In this work, the authors have structurally characterized in near-atomic resolution, the actinobacterial 2-oxoglutarate dehydrogenase complex. To this purpose, they have employed both x-ray crystallography and single-particle cryo-EM reconstructions to derive complementary insights into the complex's 3D organization, side-chain conformations, substrate placement and domain flexibility, revealing intriguing and novel information about the complex's structure.

The manuscript is very well written and extremely detailed in its presentation of all the different structural insights that have been derived during this work. Additionally, the authors have provided full validation reports for all of their reconstructions, something that even though should be mandatory, it is not, but is very appreciated. The reports further corroborate the very high quality of their presented data.

This reviewer is a bit intrigued by the choice of journal, as the manuscript feels like a better fit for a journal that is more targeted towards the structural biology community, as, for example, Nature Structural and Molecular Biology. The high-level and detailed structural descriptions in the text will probably confuse a reader who is not very familiar with structural biology. Despite that, the novelty contained in this paper makes it suitable for publication in Nature Communications.

We are grateful to the reviewer for taking the time of evaluating this work, and for her/his thorough analysis.

With all this said, there are some comments that this reviewer believes should be answered, in order to further improve the quality of the manuscript overall:

Major comments:

1) In the crystal structure, the authors report an alternate conformation of the KGD E1o domain N-ter segment, observed to form a short helical hairpin. They suggest that this can participate in intermolecular interactions and cite a work performed for the human E1o. In the eukaryotic E1o, the N-ter segment is disordered and is possibly contributing to the tethering of the E1o dimer to the periphery of the E2o cubic core (see recent publication by Hevler et al. 2023, Open Biology [10.1098/rsob.220363]) as the E2o N-ter lipoyl- arm lacks a peripheral subunit binding domain (in contrast to the presence of PSBD in PDHc E2). In the case of the actinobacterial E1o, as shown in Fig.1 of the manuscript, this N-ter helical hairpin is neither in the vicinity of E2o, or such a function, similar to the eukaryotic counterpart, should be required as the 2 proteins are in essence fused. Could the authors better describe how this region could participate in in-complex intermolecular interactions? Are there other proteins reported to participate in the formation of the oxoglutarate dehydrogenase complex in actinobacteria?

We thank the reviewer for raising this point. The MsKGD structure reported in our manuscript is indeed interesting in this regard, as it provides, in addition to a first snapshot of the enzyme homohexameric architecture, an experimental evidence of the N-terminal end of actinobacterial KGD/OdhA (first 40 residues) to be folded as an helical hairpin. In analogy to the situation described for eukaryotic E1o (Zhou et al., 2018, J. Biol. Chem. 293:19213-19227), further supported by the recent publication by Hevler et al. mentioned by the reviewer (and now included within the manuscript references), we hypothesized that the hairpin could be involved in protein-protein interactions within the mixed PDH/ODH supercomplex, possibly by tethering the enzyme to Dlat/AceF (E2p), which provides the necessary lipoyl groups. Although we agree with the referee that the hairpin does not bind to the E2o domain in the full-length MsKGD structure, we would rather caution against inferring functional information about the identity of the hairpin tethering 'target' from the sole basis of this crystal structure.

OdhA is predicted to be part of a mixed PDH/ODH supercomplex that includes, in addition to OdhA, the three components of the PDH complex (E1p, E2p, E3). Such hypothesis, first put forward by the copurification, in *C. glutamicum*, of OdhA with AceF and Lpd (E3) in pull-down experiments with Strep-tagged AceE (E1p), and vice versa with Strep-tagged OdhA (Niebisch et al., 2006, J. Biol. Chem. 281: 12300-12307), has then been supported by increasing experimental evidence, as summarized in the discussion. No other proteins have, so far, been described as part of this mixed supercomplex, although the possible presence of further, yet unknown components cannot be ruled out. The characterization of protein-protein interactions within the supercomplex, and the possible interactions of its components with other cellular structures goes however well beyond the scope of this manuscript.

2) Even though the authors have captured in their structures bound substrates in the active sites of the complex, they have not performed any additional activity assays. In this reviewer's opinion, it is critical to show that the presented complex is active, and to this purpose activity assays for the complete oxoglutarate dehydrogenase reaction. In the text (line 112-113) they mentioned activity results for the truncated, E1 domain that they have previously published. Again, in lines 181-182 they cite they previous work on E1o for the same purpose, to define the early/late stages of E1o conformations, which they again observe here.

Similar work should be performed with the complete OdhA protein to verify that the presented assembly is truly an active state and they do not only observe isolated conformations imposed only by E1o activity. This is also in

connection to the previous comment (1), as no proof is given in the manuscript that OdhA is the sole component required for the production of succinyl-CoA. Showing that the complex is truly active and can perform the complete reaction would be critical also as verification that this assembly is a true native state.

We thank the reviewer for raising this point. First, we would like to clarify that the specific decarboxylase activity reported in the text (110.3 ± 1.0 nmol/min/mg) corresponds to so-far unpublished measurements made on full-length OdhA by a DCPIP reduction assays, which are consistent with previous measurements by Hoffelder et al. (2010, J. Bacteriol. 192: 5203-5211) and with our previous measurements made on the E1-truncated construct of MsKGD (Wagner et al., 2011, Chem. Biol. 18: 1011-1020). Nonetheless, following the reviewer's suggestion, we now added both oxoglutarate dehydrogenase (ODH) and pyruvate dehydrogenase (PDH) activity measurements made on an *in vitro* reconstituted PDH/ODH complex, in which OdhA has been mixed to equimolar ratios of AceE (E1p), AceF (E2p) and Lpd (E3), all produced recombinantly. Under such conditions we do detect significant ODH activity with generation of NADH (68.6 ± 0.6 nmol/min/mg); a similar approach has also been followed in an independent study (Kinugawa et al., 2020, Microbiol. Open 9: e1113). Interestingly, the lower specific PDH activity shown by the same sample (3.3 ± 0.3 nmol/min/mg) raised to 14.5 ± 0.4 nmol/min/mg when OdhA was omitted, suggesting that OdhA and AceE may compete for the available lipoyl groups, consistently with the literature evidence showing that AceF is required for both the PDH and ODH reactions (Hoffelder et al., 2010, J. Bacteriol. 192: 5203-5211).

We otherwise agree with the reviewer that showing that OdhA is responsible for carrying out both the ThDP-dependent 2-oxoglutarate decarboxylation (E1o reaction) and the dihydrolipoamide succinyltransferase reaction (E2o) is a key point. This was already elegantly demonstrated by Hoffelder et al. in *C. glutamicum*, who, in addition to characterizing pyruvate and oxoglutarate decarboxylase activities of AceE (E1p) and OdhA, respectively, showed that OdhA does possess an additional succinyltransferase activity, activity lost by mutants in the His316/Gln320 dyad of the E2o domain. The involvement of these residues in the acyltransferase reaction is fully supported by our structural data, as illustrated in Suppl. Fig. 16. The work by Hoffelder was followed, one year later, by our study on mycobacterial KGD in which we showed that the enzyme is responsible for ODH activity (Wagner et al., 2011, Chem. Biol. 18: 1011-1020).

3) The authors performed 3DVA in order to observe the domain movements within the complex and then discuss the implications of this movement mostly in regards to the inhibition mechanism of OdhA. In regards to the "Interactions between the two catalytic centres" section, where again 3DVA is employed to describe coordination between the catalytic activities, the authors extensively describe the observed contact network but at the end of the section (lines 294-295) only briefly mention that this type of domain movement-derived contact remodeling would be critical for the complex's regulation. The authors circle the subject, but do not provide at least a hypothesis (based on their observations) about reaction intermediate channeling, given an apparent lack of a lipoyl- arm that will move reaction intermediates from the E1o domain to the E2o. Do they believe that, due to the close distance of the respective active sites, substrates only diffuse? Does the full N-ter OdhA segment (~100 residues) contain a sequence similar to the eukaryotic lipoyl- domain? Have they tried to model it with AlphaFold2? Could they elaborate on the topic?

Our considerations about the dynamic contact network between the E2o and E1o domains in OdhA are not only based on the relative domain movements suggested by 3DVA, but also on our previous work on MsKGD, in which we showed that the same kind of interdomain interactions (mediated by Arg781) were responsible, at least in part, for imposing restraints on the E1o domain (Wagner et al., 2011, Chem. Biol. 18: 1011-1020). According to the ensemble of evidence available, the channeling of the decarboxylated substrate from the E1o active site to E2o requires however the presence of lipoyl groups provided by E2p, as shown both in *Corynebacterium* (Hoffelder et al., 2010, J. Bacteriol. 192: 5203-5211) and in mycobacteria (Wagner et al., 2011, Chem. Biol. 18: 1011-1020). OdhA does not bear any lipoyl domain in its N-terminal 100-residue segment, as predicted by sequence analysis and confirmed by the AlphaFold2 model of the full-length monomer (available at <https://alphafold.ebi.ac.uk/entry/Q8NRC3>). The model shows a helical hairpin at the N-terminus of the protein, encompassing residues 8-39 and equivalent to the one observed in the MsKGD-GarA complex (Fig. 1), while the rest of the segment, characterized by a very low pLDDT, is predicted to be largely disordered. Please also see our reply to reviewer #1's comments.

4) Are the authors certain that in lower thresholds there is no density for the OdhA N-ter? Additionally, have they observed asymmetric densities in C1 reconstructions of the complex? Given enough particle abundance in their dataset, a C1 reconstruction may provide insights into asymmetric features of the complex, which, if existing, this reviewer feels they are worth reporting.

Map reconstructions in C1 were attempted for all the reported datasets, but did not provide any significant insight into possible asymmetric features of the OdhA hexamer. For this reason, they are not reported in this manuscript. In particular, no density attributable to the OdhA N-terminal harpin could be detected in any of our cryo-EM datasets, suggesting that the binding mode of the N-terminal helical segment, as observed in the MsKGD-GarA complex, may have been the result of stabilization induced by crystal packing.

5) Have the authors tried to also align the GarA with the lipoyl- domain of eukaryotic E2o? Their overall folds appear quite similar.

Similarities between the FHA domain of GarA/OdhI and lipoyl domains of known structure, regardless of the substrate specificity of the E2 enzyme they belong to, are limited to both domains being structured around a β -sandwich, with loops connecting the β -strands involved in protein-protein interactions (GarA/OdhI), or bearing the conserved, lipoyl group-carrying lysine residue (lipoyl domain). Although intriguing, there are no indications of GarA/OdhI binding possibly being competitive with respect to lipoyl-domain binding, and our high-resolution OdhA-OdhI complex argues against this hypothesis. According to the ensemble of structural and biochemical data available, GarA exerts its inhibitory effect by stabilizing the E1o domain in the resting conformation, consistently with previous observations both by us (Wagner *et al.*, 2019, *Sci. Signal.* 12: eeav9504) and by others (Balakrishnan *et al.*, 2013, *J. Biol. Chem.* 288: 21688-21702).

6) *In the discussion (lines 361-362) the authors speculate on the assembly's capability of interaction with other cellular structures. Can they suggest what those may be? Are there mentions in the literature concerning the specific topic?*

At the present state, considering the lack of literature evidence pointing to specific interactions between the PDH/ODH complex and other cellular structures in Actinobacteria, any hypothesis would be highly speculative. As summarized in our answer to the point #1, this topic is currently under investigation.

Other comments:

7) *The figures, even though quite beautiful and of very high resolution, are often confusing. Additionally, maybe some more panels should be added to increase readability, so that the reader can more closely follow the detailed structural descriptions of the text.*

We would like to thank the reviewer for her/his suggestions to improve the quality/readability of the manuscript illustrations.

In more detail:

a. *Suppl. Fig. 1: GarA domains should be colored differently for the MsKGD-GarA (orange) complex for better visibility.*

GarA chains are now colored blue for improved visibility, and GarA is also now indicated in Suppl. Fig. 1b.

b. *Fig. 1: There should be an additional panel with the N-ter helical segment zoomed in. At this state it is quite hard to observe it well.*

Two panels have been added to Fig. 1b, zooming on the N-terminal helical hairpin.

c. *Suppl. Fig. 4: Add a sub-panel showing the map density covering the β -sheet.*

A panel has been added (now as Suppl. Fig. 4c) to show 2Fo-Fc electron density for the mixed β -sheet.

d. *Fig. 2: Sub-panels should be marked with A, B, C, D and cited appropriately in the text.*

Sub-panels are now marked with letters a, b, c, d (clockwise), according to the order they are referred to in the text.

e. *Fig. 2: Show an additional sub-panel with more detail into the contacts of the short α -helix (lines 134-135).*

Instead of inserting an additional sub-panel, we rather replaced the original panel, which showed the position of this α -helix at the E2o/E1o domain interface, by a new one in which the inside face of the E2o homotrimer is shown along the 3-fold axis, thus providing a much more detailed view of intra and intersubunit interactions made by the same α -helix. The results section has also been updated with a description of these contacts. In addition, the information provided by this subpanel is complemented by Suppl. Fig. 4 (d/e/f) which shows the relative position of the α -helix within the OdhA homohexamer. Suppl. Fig. 4e, in particular, is similar to the previous top left sub-panel in Fig. 2.

f. *Add (probably in sup. Fig. 9) a panel with side-chain density coverage examples to demonstrate visually lines 168-171.*

For clarity, we rather opted to introduce a new supplementary figure (now Fig. S10) to show a representative EM map, more precisely a map section from the OdhA-succinyl-CoA complex, centered at the E2o CoA-binding site. The same map section is shown with and without the corresponding atomic model.

g. *Suppl. Fig. 16: Use different color, or better use a box to highlight conserved residues below the alignment, they get lost among the other red-colored features.*

Boxes have been added to highlight the residues cited in the text, colored according to the same coloring scheme used in the sequence logo (boxes around residues Asp798 and Ser802, not conserved, are in black). The triangles and residues id/number on the bottom of the sequence logo have also been colored accordingly.

h. Fig. 5a (line 228): Only shows the map/model generated in this paper, no comparison. It would be useful to add it. Otherwise, Sup. Fig. 1 should be cited here (it shows a general alignment).

We believe that superimposing the MskGD-GarA model (shown in Fig. 1b) to the OdhA-OdhI model shown in Fig. 5a would be detrimental to the interpretation of Fig. 5a, whose purpose is to show the overall OdhA-OdhI complex and the model to map fit. Suppl. Fig. 1 actually shows a superimposition between the MskGD-GarA complex and the previously published MskGD_{A360}-GarA complex (Wagner *et al.*, 2019, *Sci. Signal.* 12: eeav9504), obtained with an MskGD construct limited to the E1o domain. We therefore opted for adding a reference to Fig. 1b, considering that similarities and differences between the OdhA-OdhI complex and MskGD-GarA are already shown by Suppl. Fig. 18 (former Suppl. Fig. 17), and are discussed in detail in the “FHA regulation: specific interactions for a conserved inhibition mechanism” sections of the manuscript results.

i. Fig. 5b,c: Authors should consider splitting these figures and lower the amount of simultaneous side-chains displayed in each panel, in accordance with the text. The figure panel is quite small and it is hard to easily locate each side-chain discussed in-text.

We agree with the reviewer that the high number of depicted side-chains, initially intended to show the ensemble of OdhA-OdhI polar interactions, could have been detrimental to figure interpretation. We therefore followed the reviewer’s suggestion reducing the number of depicted side-chains to represent interactions mentioned in the main text.

j. Materials and Methods (lines 422-423): Authors should add a sup. fig. with SDS gels and elution profile annotating fractions.

An additional supplementary figure (now Fig. S20) has been added to provide this information.

8) Lines 117-119 (OdhA presenting... OdhA model): Should probably be rewritten to start with: “As OdhA indeed presents the same.....”.

Rephrased according to the reviewer’s suggestion.

9) Line 164: (fig 3c) in bold.

Fixed.

10) Line 173: (fig. 1b) in bold.

Fixed.

11) Line 339: stands out, not off.

Fixed.

12) Line 347: whose, not which.

Fixed.

13) Throughout the text: some words have probably been auto-corrected during writing of the manuscript to their French spelling. Inspect the text again for such occurrences.

We carefully checked the entire text for spelling and typos, and fixed where needed.

14) Line 531: state rotor speed in xg, not RPM.

In analytical ultracentrifugation velocity experiments, it is current practice to set up experiments reporting the rotational speed in RPM, since sedimentation is detected by measuring the concentration profile of the sample at different distances from the rotor center. Hence, centrifugal forces vary along the sedimentation pathway and change significantly over the course of the experiment. In contrast, the rotational speed remains constant, and, when supplied with the reference to the rotor used, provides a direct, easier indication to reproduce the experiment.

15) Line 545: L-INS-I, not linsi.

Fixed.

Reviewer #3 (Remarks to the Author):

This manuscript describes direct, structural observation of 2-oxoglutarate dehydrogenase, from an actinobacterial species. As the first structural example of such a naturally occurring, fused system version in which two component enzymes reside on a single chain, the results are novel and may be representative of this subset. In the more frequently occurring versions the multiple enzymes interact only through substrate channeling via a swinging arm mechanism. The results can be important for this class, and the experiments were appropriately carried out, described, and the results reasonably interpreted. I would however, recommend a few changes/clarifications.

We would like to thank the reviewer for evaluating this manuscript.

Line 29, "folds" should be "assembles", as it involves multiple chains.

Changed following the reviewer's suggestion.

Line 50, lipoyl domains also enter E3 to restore the initial conditions which enables cycling. Thus I would change "E1 and E2" to "E1, E2, and E3" active sites.

We agree with the reviewer's considerations and have fixed this sentence accordingly.

Lines 59 & 60, stoichiometries and interactions are NOT elusive for most TETHERING (psbd) interactions, as they have been determined, visually observed, and reported. Only lipoyl domain interactions with the peripheral components remain elusive.

We agree with the reviewer that several past works have been focused on the structural characterization of PSBD domains from different sources, and a few structures of E1-PSBD and E3-PSBD complexes are available. We therefore rephrased the sentence to cite such works (now as references # 15-20), and rather state that "the intrinsic flexible nature of the E2 swinging arms has long hampered attempts to perform high-resolution structural studies of these complexes in their entirety."

Lines 60 & 61, this is misleading since only CORE component E2 or E3 binding protein trimer interactions, and mainly those forming the pentamers, were visible; NO channeling interactions, which MUST involve the peripheral E1 and E3 components, were EXPERIMENTALLY observed in those studies.

Although we agree with the reviewer that channeling interactions involving the E1 or E3 components have yet to be experimentally observed, the 2021 study by Skerlová *et al.* (Nature Commun. 12: 5277) reports a 3.16 Å cryo-EM structure of E2 inner core of *E. coli* PDH, with bound lipoyl domains, and details about the insertion of dihydrolipoyl-lysine into the E2p active site are provided (PDB 7B9K). To avoid confusion, we therefore rephrased the sentence to specify "single particle cryo-EM studies that have shed light on lipoamide channeling within the E2 core of *E. coli* PDH²¹, as well as on the role of the E3 binding protein ...".

Line 68, I would suggest "two-in-on" rather than "all-in-one" here, in the title, and elsewhere, since activities associated with E3 (ring closure, NADH production/release) are not present, and the paper also speculates on the use of lipoyl domains from PDH in a mixed complex to enable acyl transfer. Clearly the reported complex described can not do it "all."

We thank the reviewer for this suggestion and replaced all occurrences of "all-in-one" by "two-in-one", starting from the manuscript's title.

Does production in E. Coli affect assembly of the non-E. Coli complex, given that a related version is also present naturally? This should be mentioned.

Although the formation of 'mixed' complexes inside *E. coli* cells cannot be ruled out, and the presence of bound acetyl-CoA to OdhA is likely to be ascribed to intracellular conditions during overexpression in *E. coli*, no significant cross-contamination by the corresponding, native *E. coli* enzymes have been detected in our recombinant proteins. The nature of the OdhA homohexamer, in which each domain keeps its obligate oligomerization state while keeping close interactions related to enzyme regulation (see above), the stability of the homohexameric state in solution (Suppl. Fig. 2) and the observation of such architecture being conserved in the mycobacterial counterpart, argue against an effect induced by possible interactions with *E. coli* native enzymes.

Fig 1 caption should state the model resolution. The citing text indicates 4.6 angstroms, although electron density is not shown in the figure.

The model resolution is now specified in Fig. 1 caption. Details about data collection, models refinement and validation are provided in Tables 1 and 2.

Line 142. I think “Fig 2” should be “Fig 4” or the wording changed as Fig 2 shows no electron density.

We reworded the reference to this figure, which now points more specifically to Fig. 2c, stating “Consistently, we could model CoASH, which was added to the cocrystallization mixture, as bound to the E2o acceptor site (Fig. 2c)...”.

Lines 328-329, It can be argued that features enabling substrate channeling are still largely unknown since transient lipoyl domain recognition/binding has not been reconciled, especially with E1 and E3 peripheral components. The recent findings cited are confined to the cores, and so far ALL experimental methods, including cryoEM have been dismal failures regarding channeling, which MUST involve the other components. I would tone this down. There are still many remaining issues to be resolved!

We agree with the reviewer’s considerations, and rephrased the sentence as “..the last couple of years have however seen significant advances in the field, ranging from the composition of eukaryotic E2p and E2o cores^{22-24,43-45}, to snapshots of protein interactions and dihydrolipoyl-lysine entering the E2 active site^{2,21,24,46}.”

Line 332 what was “cutting edge”? I didn’t catch any special sub-particle optimization or new treatment as it seemed to be routine class-averaging.

We removed the ‘cutting-edge’ expression from this sentence.

Lines 356-359. How do you know PDH and ODH are in physical proximity in cells? Co-purification can be the result of assembly after cell lysis. Has a mixed complex ever been observed, and if so, what’s its size and stoichiometry? This should be discussed.

We thank the reviewer for this comment. Current evidence supporting physical proximity of MsKGD/OdhA to the PDH components relies on the ODH reaction dependence on AceF (E2p), in addition to co-purification studies from *C. glutamicum* reported by at least two independent groups (Niebisch *et al.*, 2006, J. Biol. Chem. 281: 12300-12307; Kinugawa *et al.*, 2020, Microbiol Open 9: e1113). Direct experimental evidence of the nature of such a complex *in vivo*, and a characterization of its overall properties is however yet to be provided and constitutes a topic which is actively investigated. We believe that the present unknowns on the nature of this complex are already mentioned in the concluding section of this manuscript, when we state “The next challenging goals will include determining how OdhA and the other components of the PDH/ODH supercomplex may be physically and temporally assembled in a supramolecular structure, and whether such an assembly could interact with other cellular structures”. We would also like to refer the reviewer to our discussion to comment #1 by reviewer #2.

REVIEWERS' COMMENTS

Reviewer #2 (Remarks to the Author):

This reviewer is satisfied with answers to all their comments but for major comment 2. In addition, answers of the authors provided intriguing ideas regarding their results, please see below:

Major comment 2:

The inclusion of high-quality kinetic data is absolutely crucial for a comprehensive discussion on bound substrates and released products in this manuscript. At present, there is a disconnection between the newly incorporated kinetic data and the structurally captured substrates. The paragraph added in the manuscript only presents corresponding activities in a preliminary manner, lacking analysis, structure/function correlations and/or specific activity plots. The supplementary table only includes the plate reader data with no extra analysis - a main figure or, at least panels with standard deviations, number of data points, replicates etc would be appreciated. Furthermore, it is unclear if current measurements represent technical or biological triplicates or for which exact construct they were measured. Hence, I strongly encourage the authors to conduct a deeper and more consistent kinetic analysis of their constructs.

Other Major comments:

A-I have realized that a mechanistic model for the overall function is missing and all these beautiful data and results from the authors must be connected to a final figure mechanistic model for OGH.

B-The cryo-EM data would be definitely useful as open-source in EMPIAR. Please upload them there.

My only minor point is to please mention explicitly in the manuscript which are crystal or cryo-em structures (especially in the figure legends), as this reader was confused while re-reading the manuscript.

Provided the above are included in the manuscript, I am eager to accept the manuscript for publication.

Reviewer #3 (Remarks to the Author):

The manuscript has been revised appropriately with respect to my and other reviewers' previous comments. It describes an important contribution to the field, and I now have only two, very simple suggestions.

1) on line 45, I would add after "PDH", (PDHc). including the comma and parentheses, as PDHc is how it's often been referred to for many years now. FYI the trailing "c" designates "complex", whereas PDH is sometimes confusing as it's been used at times, to mean both the entire complex, and the pyruvate decarboxylating E1 component.

2) on line 57, "connected" should be "covalently connected"

Yang, L., Wagner, T., Mechaly, A.M., Boyko, A., Bruch, E.M., Megrian, D., Gubellini, F., Alzari, P.M., Bellinzoni, M.

High resolution cryo-EM and crystallographic snapshots of the actinobacterial two-in-one 2-oxoglutarate dehydrogenase (updated title)

REPLIES TO REVIEWER COMMENTS

Reviewer #2 (Remarks to the Author):

This reviewer is satisfied with answers to all their comments but for major comment 2.

We would like to thank the reviewer for evaluating the revised version of this manuscript, and for her/his detailed feedback.

In addition, answers of the authors provided intriguing ideas regarding their results, please see below:

Major comment 2:

The inclusion of high-quality kinetic data is absolutely crucial for a comprehensive discussion on bound substrates and released products in this manuscript. At present, there is a disconnection between the newly incorporated kinetic data and the structurally captured substrates. The paragraph added in the manuscript only presents corresponding activities in a preliminary manner, lacking analysis, structure/function correlations and/or specific activity plots.

Enzymatic activity data that were added to the revised version of the manuscript consist in specific PDH and ODH activity measurements on the *in vitro* reconstituted PDH/ODH supercomplex, as requested after the first round of review. The purpose for the inclusion of these data was to provide, as requested by the reviewer, a quality control of recombinant OdhA to check that the observed assembly corresponds to an active state. Not only the OdhA constructs are active, but the relation between activity and the conformational states of the enzyme is largely supported by the ensemble of our structural observations. These points are extensively discussed in the manuscript with references to previous work on the mycobacterial homologues, both from us on *MsKGD* (Wagner *et al.*, 2011, *Chem. Biol.* 18: 1011-1020; Wagner *et al.*, 2014, *Biochem. J.* 457: 425-434; Wagner *et al.*, 2019, *J. Struct. Biol.* 208: 182-190) and from others (Balakrishnan *et al.*, 2013, *J. Biol. Chem.* 288, 21688-21702). On top of that, it should be noted that a thorough enzymatic characterization of *C. glutamicum* OdhA has been published by L. Eggeling's group back in 2010 (Hoffelder *et al.*, 2010, *J. Bacteriol.* 192, 5203-5211). In that work, the authors showed both OdhA's capability to perform the ThDP-dependent, oxidative 2-oxoglutarate decarboxylation (E1o) and the synthesis of succinyl-CoA (E2o), the latter shown to depend on AceF (E2p) for supplying the essential lipoyl group, confirming earlier hypotheses (Niebisch *et al.*, 2006, *J. Biol. Chem.* 281: 12300-12307). Those results found further confirmation in a more recent work (Kinugawa *et al.*, 2020, *Microbiologyopen* 9: e1113) that reported ODH specific activity measurements and characterization of an *in vitro* reconstituted PDH/ODH supercomplex from *C. glutamicum*. In addition, Hoffelder *et al.* performed steady-state characterization of *C. glutamicum* OdhA (providing V_{max} and K_m values for 2-oxoglutarate, among other data) and reported the effect of point mutations on residues involved in CoA binding. Those findings are fully consistent with our work and are cited several times throughout the manuscript for structure/function considerations on the bound substrates/products.

The supplementary table only includes the plate reader data with no extra analysis - a main figure or, at least panels with standard deviations, number of data points, replicates etc would be appreciated. Furthermore, it is unclear if current measurements represent technical or biological triplicates or for which exact construct they were measured.

The supplementary table which is being referred here by the Reviewer is actually the source data file, that we provided with the revised version according to the journal's editorial requirements. The provided raw data refer to technical replicates of either decarboxylase activity measurements (DCPIP assay) of ODH or PDH activity measurements for the *in vitro* reconstituted PDH/ODH complex, as indicated in the respective excel worksheets.

Nevertheless, for a clearer presentation of our data and an easier reference to previous literature on the topic, we have included a supplementary item (Supplementary Table 1) which shows our values in comparison with similar findings published elsewhere.

Hence, I strongly encourage the authors to conduct a deeper and more consistent kinetic analysis of their constructs.

We believe that performing further kinetic experiments on either *MsKGD* or OdhA, which would take significant time and resources, would go well beyond the scope of this work and would, most likely, fall short in providing

significant new insights, with respect to both the findings presented in this work and previously published biochemical work on these enzymes. Please see our comment above.

Other Major comments:

A-I have realized that a mechanistic model for the overall function is missing and all these beautiful data and results from the authors must be connected to a final figure mechanistic model for OGH.

We would like to thank the reviewer for this suggestion, that we followed introducing a concluding figure in the main manuscript depicting a simplified, mechanistic model for the whole ODH reaction with a focus on the reactions catalyzed by either OdhA domain (Figure 7).

B-The cryo-EM data would be definitely useful as open-source in EMPIAR. Please upload them there.

We also agree with the reviewer on this point and started the deposition of relevant raw datasets to the EMPIAR database. Upload of the data, which span several thousand files (microscopy 'movies') per dataset, is currently in progress considering bandwidth and other technical limitations linked to the amount of data to be transferred (~20 Tb overall). We will be able to finalize them and release the whole raw data concomitantly the manuscript publication and the release of the respective PDB and EMDB entries.

My only minor point is to please mention explicitly in the manuscript which are crystal or cryo-em structures (especially in the figure legends), as this reader was confused while re-reading the manuscript.

Figure legends now mention whether models were obtained by X-ray crystallography or single particle cryo-EM.

Provided the above are included in the manuscript, i am eager to accept the manuscript for publication.

Reviewer #3 (Remarks to the Author):

The manuscript has been revised appropriately with respect to my and other reviewers' previous comments. It describes an important contribution to the field, and I now have only two, very simple suggestions.

We would like to thank the reviewer for evaluating this revised version and for her/his new suggestions.

1) on line 45, I would add after "PDH", (PDHc). including the comma and parentheses, as PDHc is how it's often been referred to for many years now. FYI the trailing "c" designates "complex", whereas PDH is sometimes confusing as it's been used at times, to mean both the entire complex, and the pyruvate decarboxylating E1 component.

We changed "PDH" to "PDHc" as suggested by the reviewer (now line 36) and shortly after (line 61), whereas we left the acronym "PDH" elsewhere in the manuscript, as the remaining citations were either followed by the word "complex" (singular or plural) or explicitly referring to pyruvate dehydrogenase as the catalyzed reaction, instead of the protein complex.

2) on line 57, "connected" should be "covalently connected"

Changed as suggested (now on line 48).